# Are there lane advantages in track and field?

**David Munro** *

Economics Department, Middlebury College, Middlebury, VT, United States of America

* dmunro@middlebury.edu

## Abstract

Shorter distance events in track and field are replete with folk tales about which lane assignments on the track are advantageous. Estimating the causal effect of lane assignments on race times is a difficult task as lane assignments are typically non-random. To estimate these effects I exploit a random assignment rule for the first round of races in short distance events. Using twenty years of data from the IAAF world athletic championships and U20 world championships, there is no evidence of lane advantages in the 100m. Contrary to popular belief, the data suggest that outside lanes in the 200m and 400m produce faster race times. In the 800m, which is unique in having a lane break, there is some weak evidence that outside lanes producer slower race times, possibly reflecting the advantage of inside lanes having an established position on the track at the lane break. Given that these results do not support common convictions on lane advantages, they also serve as an interesting case study on false beliefs.

**Data Availability Statement:** All data is publicly available and can be accessed via: https://www.worldathletics.org/competitions A full replication package is included in my submission materials and, in addition, can be located here: https://github.com/dmunro-git/Lane-Advantages.

## Introduction

In shorter distance track and field events one frequently encounters tales about lane advantages, that is, which lane assignments on the track produce the fastest event times. This track and field folklore is often heard from coaches, teammates, etc., however they are also codified in competition rules, and appear in the popular press (e.g. [1, 2]). These beliefs are held at the highest levels of the sport. Following his bronze medal win in the 100m at the 2020 Olympics Andre De Grasse noted: "I knew it was going to be a tough one after I drew lane nine. I didn't have a great semifinal and I knew I had to come out and try and execute as best as I can" [3]. In the context of this paper, readers may find it interesting to note that De Grasse's race in lane nine was his personal record.

Common narratives claim that in races with corners, running in the outside lane (typically 7 through 8 or 9) is disadvantageous as the runner cannot see any of their competitors, and that the very inside lanes (typically 1 and 2) are also seen as undesirable as they have the tightest corners. Therefore, the middle lanes of the track are deemed the most desirable. In support of the belief that inside corners are slower, researchers examining the biomechanics of running find evidence that tighter corners do in fact slow runners down. Tighter corners both reduce running speeds (e.g. [4, 5]) and have lower foot force production [6]. However, no such empirical evidence exists to support the claim that one's inability to see competitors when running in an outside lane creates a disadvantage. While beliefs about lane advantages are commonly

**Funding:** The author received no specific funding for this work.

**Competing interests:** The authors have declared that no competing interests exist.

connected to races with corners, the De Grasse quote above highlights that these beliefs persist in events run on straightaways. There is also no empirical work examining the existence of lane advantages in straightaway races. While assessing the impact of seeing competitors, and its ultimate effect on race times, is clearly a relevant question in the context of track and field performance, it also relates more broadly to questions about the performance effects of motivational or psychological factors. For example, there is some evidence [7] that sports teams losing games at halftime end up winning the game more frequently (though [8], find opposing evidence), that professional golfers respond differently to the possibility of losses [9], and, in the non-sports world, that incentives framed as losses (as opposed to gains) can increase worker productivity [10]. This paper contributes to this literature by analyzing these motivation or psychological effects in a track and field context.

Estimating the causal impact of these lane advantages is a difficult empirical task as lane assignments are typically non-random and instead are a function of seed times or race times in prior elimination rounds of the event. Lanes deemed as advantageous in the folklore are assigned to runners with superior seed times or qualifying times. This endogenous assignment to treatment (lanes) prevents a causal interpretation of any differences observed in race times by lane.

Besides the biomechanical evidence, other researchers have approached this question in various ways. Using a mathematical model of track geometry, [11] exploit different track designs and find tighter corners (smaller radii) produce slower times. [12] examines the effects that lane assignments on world records, but does not control for endogenous assignment to lanes. [6] examines the effects of lane assignments on placings in races, and only find statistical differences in events with endogenous assignment to lanes.

To overcome the issue of endogenous assignment to lanes and obtain a causal estimate of the effect of lane assignments on race times, I exploit a random assignment rule used in the first round of major track meets. Using twenty years of data from the IAAF, I estimate these effects in the 100m, 200m, 400m, and 800m. Beginning with the 100m, the data suggests that lane assignments have no effect on race times. This null effect is precisely measured and, given the level of statistical power in the analysis, if true lane advantages do exist in the 100m, the precise null results suggest they must be quite small. In the 200m, there is robust evidence that outside lanes on the track produce the fastest race times. Further, average race times appear to be roughly monotonically decreasing with lane number. This is consistent with the evidence on the biomechanics of running, but inconsistent with the view that outside lanes are undesirable. The common belief in the track and field folklore is that the middle lanes (often 3–6) are the most desirable. Thus, these estimates suggest that these beliefs are incorrect. To give a brief sense of magnitude, depending on the estimation strategy, I find that lane 8 produces, on average, race times which are between 0.084 and 0.178 seconds faster than lane 2. While this is small in absolute magnitude, it could amount to important differences in race results given the standard deviation of race times in the data is less than 1 second.

Consistent with the results in the 200m, outside lanes in the 400m also appear to produce faster race times on average. Though these results are somewhat weaker statistically than those from the 200m. It is important to emphasize that as average race times increase so do the dispersion of race times, so statistical power becomes more of an issue in the 400m and 800m. In an alternative estimation approach which pools runners to increase statistical power, there is stronger statistical evidence that outside lanes in the 400m are faster on average and the magnitude of the effects are similar to those found in the 200m. Finally, in the 800m, there is some mixed evidence that outside lanes produce slower race times. The 800m is unique in this collection of events in its use of a lane break. Thus, the result that outside lanes produce slower times is consistent with the notion that inside lanes are advantageous in the 800m as those

runners have an established position on the inside of the track at the lane break. Results from all these events are generally consistent across various statistical models.

These baseline results reflect the *net* impact of lane assignments where both the tighter corners (biomechanical effects) and the motivational or psychological effects related to seeing competitors could be simultaneously impacting race times. The evidence from the 200m highlights that if the motivational/psychological effects do slow runners down in outside lanes, they are dominated by the tight corners effect. Leveraging the fact that the outermost lane on the track (where runners see no other runners for some portion of the race in the 200 and 400m) are not perfectly correlated with lane number, I also estimate the *marginal* effect of being in the outermost lane. In the 200m, there is evidence that, all else equal, being in the outermost lane slows runners down. This is suggestive that race times may be influenced to some degree by motivational or psychological factors related to seeing competitors.

I end the paper with some discussion of the implications of these results for race rules, and address why common beliefs about lane advantages are not supported by the data. Finally, I also highlight what other events or competitions this approach to estimating the effects of lane assignments could be implemented.

## Data and empirical strategy

The data come from IAAF World Championships and U20 World Championships from 2000 to 2019 and was accessed from [13]. Prior to 2000, World Championship data did not include reaction times for the 100m through 400m, and data on season's bests and personal bests, which are important regressors below, become very sparse. As a result, I focus on post-2000 data. Data was collected for Men's and Women's 100m, 200m, 400m, and 800m. In aggregate, this amounts to roughly 8000 individual race times for these events over this time period. A replication package for the analyses conducted in this paper can be found here: [14]

### Causal inference framework

The causal effect of lane assignments involves estimating how a runners performance would have changed if they ran their race in a different lane. Denote runner $i$ in heat $j$ observed race time as $Y_{i,j}$. There are typically nine lanes on the track, and so possibly nine treatment statuses, but as a simple illustrative example, suppose we are interested in measuring the causal effect of running in lane 8. One lane must be chosen as a reference point to compare all other lanes against, and as is discussed below, I choose lane 2 for this. Denote $T_{8,i,j}$ as a binary indicator variable denoting assignment to lane 8: $T_{8,i,j} = \{0, 1\}$. The observed race time can be written in terms of potential outcomes:

$$Y_{i,j} = \begin{cases} Y_{8,i,j} & \text{if} \quad T_{8,i,j} = 1 \\ Y_{2,i,j} & \text{if} \quad \sum_{\substack{1 \leq k \leq 9 \\ k \neq 2}} T_{k,i,j} = 0 \end{cases} \tag{1}$$

Where the last condition is when all other indicator variables are zero, and the runner is in lane 2. In the lane 8 example,

$$Y_{i,j} = Y_{2,i,j} + (Y_{8,i,j} - Y_{2,i,j})T_{8,i,j} \tag{2}$$

$(Y_{8,i,j} - Y_{2,i,j})$ is the causal effect of running in lane 8 (relative to lane 2). The fundamental empirical challenge in estimating the true causal effect of lane assignments on race times is that, for most races, the assignment to lanes is conditional on a runner's ability. To understand this issue in this example, the difference in average race times between lanes 2 and 8 can be

written as:

$$E[Y_{i,j}|T_{8,i,j} = 1] - \underbrace{E[Y_{i,j}|\sum_{\substack{1 \le k \le 9 \\ k \ne 2}} T_{k,i,j} = 0]}_{\text{Observed difference in average race times}} = \underbrace{E[Y_{8,i,j}|T_{8,i,j} = 1] - E[Y_{2,i,j}|T_{8,i,j} = 1]}_{\text{Avg. treatment effect on the treated}} +$$

$$\underbrace{E[Y_{2,i,j}|T_{8,i,j} = 1] - E[Y_{2,i,j}|\sum_{\substack{1 \le k \le 9 \\ k \ne 2}} T_{k,i,j} = 0]}_{\text{Selection bias}}$$

(3)

The selection effect term captures the difference in average $Y_{2,i,j}$ between those who were assigned to lanes 8 and 2. In races where assignment to lanes is not random, the selection bias term will not be equal to zero (e.g. runners assigned to lanes conditional on ability). Random assignment of lanes eliminates the selection bias term in Eq (3). With random assignment $E[Y_{2,i,j}|T_{8,i,j} = 1] = E[Y_{2,i,j}|\sum_{\substack{1 \le k \le 9 \\ k \ne 2}} T_{k,i,j} = 0]$ (independence of runner ability and treatment status), thus the selection bias term in Eq (3) cancels. For more discussion on this see [15].

To quantify the observed difference of race times across lanes I estimate the following statistical model:

$$Y_{i,j} = \alpha_0 + \sum_{\substack{1 \le k \le 9 \\ k \ne 2}} \beta_k T_{k,i,j} + \alpha_f X_{f,i,j} + \epsilon_{i,j}$$

(4)

Where $Y_{i,j}$ denotes the observed race time of runner $i$ in heat $j$, $T_{k,i,j}$ denotes indicator variables for each lane (excluding lane 2, the reference lane), and $X_{f,i,j}$ denotes a collection of control variables. Simply estimating Eq (4) on all race data would likely lead to biased estimates of treatment effects as lane assignments are typically non-random. To overcome this fundamental issue and to estimate the causal effects of lane assignments on race times, I leverage the random assignment rule implemented by the IAAF in the first round of each event. This random assignment is important from a causal inference perspective as the runners in each lane will (on average) have the same characteristics, and thus any differences in race times can be attributed to lane assignments. This is the independence criteria highlighted above. Specifically [16], states: "In the first round and any additional preliminary qualification round as per Rule 166.1, the lane order shall be drawn by lot." From personal correspondence with rules officials at the IAAF I have confirmed that this random assignment rule was initiated in the 1985–86 rulebook under rule 141.11 and is still in place today. In recent years, the Men's 100m in the World Championship also included Preliminary Round heats, which occurred prior to Round 1 heats. This preliminary round is for "unqualified" athletes. According to the rules, random assignment to lanes is supposed to occur in both the Preliminary Round and Round 1 heats. However, from examining the data, and corresponding with rules officials, it appears that the fastest race times from the Preliminary Round heats were sorted into the outside lanes for the Round 1 heats, resulting in a non-random assignment in Round 1 of these events. As such, for the events where there is a Preliminary Round prior to Round 1, I exclude the Round 1 data from the analysis.

In practice, one could explore the average differences in race times across lanes in a non-parametric manner (e.g. t-tests). However, runner ability varies quite a bit in the first heats and, as a result, there is substantial variation in race times. This makes detecting any statistical differences challenging. The use of the statistical model in Eq (4) is useful as it includes various control variables which help explain much of this variation and helps to sharpen the estimates of lane effects.

The following control variables ($X_{f,i,j}$) are included in the regression; the recorded wind measurement in each heat (which is only included in 100m and 200m events) and positive (negative) measurements denote tailwinds (headwinds), the runner's reaction time to the start gun (which is included in the 100m, 200m, and 400m), the runner's season best race time, the runner's personal best race time, and, when data for Men's and Women's races are pooled, a dummy variable indicating male events. An additional desirable feature of including season's best is that it controls for any year effects (e.g. sprinters getting faster over time). The coefficient of interest is on the lane dummy variables, which estimate the causal effect of lane assignments on race times. The additional covariates are useful in explaining much of the variation in race times, which helps to sharpen the estimates of lane effects. Another approach to estimate lane effects would be to exploit within sprinter variation (i.e. observing the same sprinter in multiple lane assignments). However, the vast majority of runners (70–80%) appear in the data only once, which severely hampers such an approach. The inclusion of personal best in regression Eq (4) plays this role to some degree for athletes who are in the data more than once, but only if personal best is not changed between observations of the same athlete.

## Results

### 100m

I begin by analyzing lane assignment effects in the 100m. Narratives about lane advantages tend to be focused on races with corners (200m and 400m) but the quote from Andre De Grasse in the introduction highlights that they also persist in the 100m. Similar to the 200m and 400m, the beliefs that middle lanes are best in the 100m could relate to the fact that middle lanes improve a runners vantage and helps them judge where they are relative to their competitors. Indeed, in lane assignment rules used for later rounds of races, the fastest qualifying times are assigned to inside lanes, which suggests they are viewed as favorable.

To begin each analysis, I confirm whether the randomization across lanes is effective. To do this I estimate the following statistical model:

$$Y_{i,j}^{SB} = \alpha_0 + \sum_{\substack{1 \leq k \leq 9 \\ k \neq 2}} \beta_k T_{k,i,j} + \alpha_f X_{f,i,j} + \epsilon_{i,j} \tag{5}$$

Where $Y_{i,j}^{SB}$ denotes a runner's season's best. If runners are assigned lanes based on ability (e.g. their performance in meets taking place earlier in the season), this would be highly problematic for assessing the causal impact of lane assignments. To qualify for the World or U20 Championships, athletes must meet the entry standard in a window that typically spans a year prior to the event. To insure that lane assignments in Round 1 are indeed random, I proxy for an athletes ability with their season's best (prior to the event being analyzed) and test whether there are statistical differences in season's best across lanes. Results from this randomization check are reported in Table 1 below. As discussed below, lane 2 was chosen as the baseline to compare against all other lanes.

Columns 1 and 3 in Table 1 report the results from estimating Eq (5) using all data from the Men's and Women's races, respectively. As discussed in more detail below, columns 2 and 4 report the results with outliers excluded. Columns 5 and 6 report the results using all data and data with outliers removed for the pooled Men's and Women's data.

In general, the randomization appears to effectively balance runners into lanes based on their season's best times. In a few cases there are statistically significant results, but these normally appear in lanes with a low number of observations, reported in square brackets. It is not reported in the table because there are no corresponding regression results, but lane 2 has a

**Table 1. Randomization check for 100m.**

| Coeff. (Ind. Var.) | Mens | | Womens | | Pooled | |
|---|---|---|---|---|---|---|
| $\beta_1$ (Lane 1) | -0.0167 | -0.0354 | 0.1576† | 0.1642** | 0.0365 | 0.0344 |
| | (0.0510) | (0.0408) | (0.1112) | (0.0751) | (0.0555) | (0.0394) |
| | [74] | [71] | [40] | [37] | [114] | [108] |
| $\beta_3$ (Lane 3) | -0.0226 | -0.0474 | 0.0175 | 0.0381 | -0.0016 | -0.0043 |
| | (0.0521) | (0.0429) | (0.0905) | (0.0682) | (0.0525) | (0.0403) |
| | [98] | [93] | [103] | [95] | [201] | [188] |
| $\beta_4$ (Lane 4) | -0.0497 | -0.0483 | -0.1836** | -0.119* | -0.1168** | -0.0838** |
| | (0.0488) | (0.0427) | (0.0857) | (0.0611) | (0.0494) | (0.0371) |
| | [104] | [101] | [104] | [101] | [208] | [202] |
| $\beta_5$ (Lane 5) | 0.0101 | -0.0133 | -0.0046 | 0.0332 | 0.0017 | -0.0090 |
| | (0.0502) | (0.0409) | (0.0857) | (0.0650) | (0.0500) | (0.0378) |
| | [111] | [105] | [101] | [95] | [212] | [201] |
| $\beta_6$ (Lane 6) | -0.0432 | -0.0627* | -0.0068 | -0.0100 | -0.0264 | -0.0372 |
| | (0.0463) | (0.0377) | (0.0929) | (0.0646) | (0.0515) | (0.0368) |
| | [110] | [106] | [104] | [97] | [214] | [203] |
| $\beta_7$ (Lane 7) | 0.0992* | 0.0474 | -0.1383* | -0.0737 | -0.0193 | -0.0149 |
| | (0.0549) | (0.0462) | (0.0834) | (0.0575) | (0.0503) | (0.0370) |
| | [105] | [96] | [104] | [101] | [209] | [197] |
| $\beta_8$ (Lane 8) | 0.0081 | -0.0391 | -0.0822 | 0.0219 | -0.0371 | -0.0080 |
| | (0.0521) | (0.0444) | (0.0785) | (0.0597) | (0.0471) | (0.0371) |
| | [98] | [90] | [97] | [95] | [195] | [185] |
| $\beta_9$ (Lane 9) | 0.0588 | 0.0487 | -0.2905*** | -0.1990** | -0.1313* | -0.0890† |
| | (0.0810) | (0.0730) | (0.1008) | (0.0823) | (0.0680) | (0.0578) |
| | [26] | [25] | [32] | [31] | [58] | [56] |
| $\alpha_1$ (Male) | | | | | -1.125*** | -1.087*** |
| | | | | | (0.0249) | (0.0194) |
| $\alpha_0$ (constant) | 10.53*** | 10.50*** | 11.71*** | 11.58*** | 11.68*** | 11.58*** |
| | (0.0321) | (0.0284) | (0.0658) | (0.0444) | (0.0414) | (0.0292) |
| N | 828 | 786 | 788 | 748 | 1616 | 1534 |
| $R^2$ | 0.0135 | 0.0136 | 0.028 | 0.0305 | 0.568 | 0.68 |
| Outliers Removed | No | Yes | No | Yes | No | Yes |
| F-stat. | 1.19 | 1.21 | 3.28 | 3.44 | 1.84 | 1.69 |
| p-value | 0.2995 | 0.2893 | 0.0011 | 0.0007 | 0.0658 | 0.0957 |

This table reports the coefficients estimated from model Eq (5). To ease interpretation of the results, the independent variable associated with each coefficient estimate is highlighted in parentheses. The number of observations per lane are reported in square brackets. Robust standard errors (see [19]) are reported in parentheses.

†, *, **, and *** denote significance at the one-sided 10%, two-sided 10%, 5%, and 1% levels, respectively.

similar number of observations to lane 3. A common issue in the 100m, and all other events, is that in these first round races lanes 1 and 9 are often empty. As an example, in Table 1 the Women's 100m has 40 or fewer observations in lanes 1 and 9, relative to around 100 observations in the other lanes. Because lanes 1 and 9 have much fewer observations than the other lanes, they are more susceptible to issues relating to low statistical power. As such, all estimated lane effects for lanes 1 and 9 throughout this paper should be treated with caution as they are more susceptible to Type-1 error (see, e.g., [17]). In addition, small sample sizes are susceptible to Type-M error (exaggerating the magnitude of the effects) [18].

At the bottom of the randomization tables, F-statistics and their associated p-values are reported for joint significance tests of the lanes. Only the women's races are jointly significance at the 5% level, and when this data is pooled with the Men's data, the lane estimates fail significance at the 5% level, suggesting that the randomization is generally effective.

As an additional robustness check, I examine if the propensity (probability) a runner is assigned to a specific lane is statistically related to their season's best. These results are reported in Tables 12–15 in S1 Appendix. None of the regressions show a significant relationship between season's best and treatment status (lane assignments), providing additional evidence that the randomization is effective.

Moving on to the estimates of the effect of lane assignments on race times, results from model Eq (4) for the 100m data are reported in Table 2. Racers who do not start (DNS) or who are disqualified (DQ) to not register race times. In addition, I exclude any racers with missing season and personal best data as these are important in the analysis. Again, I report the results separately for Men's and Women's races and also pool the Men's and Women's data in the "Pooled" columns to help improve statistical power. I chose lane 2 as the baseline to compare the other lanes against. I do this because, as is highlighted above, lane 1 consistently has much fewer observations than lanes 2 through 8 and thus may be more susceptible to issues relating to low statistical power. Columns 1 and 3 in Table 2 do not show any systematic effect of lane assignments on race times. There are a few statistically significant lane effects in the Men's data in column 1. For example, lane 3 produces race times which are on average 0.061 seconds slower than lane 2 (significant at the 5% level). Where as in the Women's data (column 3), for example, lane 7 is 0.0442 seconds faster on average relative to lane 2 (weakly significant at the 1-sided 10% level). However, the lack of consistency between lanes within the Men's and Women's races, along with the lack of consistency between genders suggests these results may be anomalous. Pooling the Men's and Women's data (column 5) to improve statistical power only yields a weakly significant effect (1-sided 10%) for lane 9 (-0.043 seconds faster than lane 2). But again, this result should be treated with caution as it has far fewer observations than other lanes.

Another concern one might have with the data is the presence of extreme outliers. For example, in the data there are race times that are more than three standard deviations slower than the mean. It is possible these extreme outliers have an important influence on the estimated lane effects. It also seems plausible that these extreme outliers are unrelated to lane assignments. For example, a runner who sustains an injury during the race may have a much slower race time than the norm. To control for these outliers, I conduct the same analysis where I exclude the slowest 5% of the race times in the Men's and Women's race, reported in columns 2 and 4 respectively, and results from the pooled analysis are reported in column 6. Excluding these extreme outliers does not generate a meaningful change in the overall regression results in the 100m. In the pooled data with outliers excluded only lane 9 again has a significant lane effect, being on average 0.0513 seconds faster than lane 2. While this effect should be treated with caution because of the low number of observations, it is also the opposite effect relative to the common narrative pertaining to outside lanes in the 100m.

An important point worth emphasizing with this empirical strategy is that runners, of course, are not blinded to their lane assignments. In an analogy from clinical trials for drugs, it is as if "control" subjects do not receive a placebo and are aware of their treatment status. A concern with non-placebo trials is that control subjects engage in differential behavior because of their status (e.g. seek their own treatment) which may impact the estimates of treatment effects. Though leveraging random assignment ensures runner characteristics will be balanced across lanes, it is possible that runners adjust their effort in response to their lane assignments. The main objective in these early rounds is to qualify to advance to later rounds. Runners may

**Table 2. Regression results for 100m.**

| Coeff. (Ind. Var.) | Mens | | Womens | | Pooled | | |
|---|---|---|---|---|---|---|---|
| $\beta_1$ (Lane 1) | 0.0194 | 0.0257 | -0.0137 | -0.0267 | 0.0030 | 0.0033 | 0.0081 |
| | (0.0232) | (0.0222) | (0.0473) | (0.0339) | (0.0238) | (0.0194) | (0.0201) |
| $\beta_3$ (Lane 3) | 0.0610** | 0.0539*** | -0.0238 | -0.0320 | 0.0223 | 0.0145 | 0.0130 |
| | (0.0255) | (0.0199) | (0.0333) | (0.0257) | (0.0210) | (0.0163) | (0.0175) |
| $\beta_4$ (Lane 4) | 0.0453† | 0.0227† | -0.0187 | -0.0059 | 0.0133 | 0.0087 | 0.0039 |
| | (0.0326) | (0.0172) | (0.0292) | (0.0251) | (0.0220) | (0.0154) | (0.0170) |
| $\beta_5$ (Lane 5) | 0.0450 | 0.0234† | -0.0264 | -0.0127 | 0.0105 | 0.0071 | 0.0118 |
| | (0.0356) | (0.0171) | (0.0308) | (0.0265) | (0.0238) | (0.0158) | (0.0170) |
| $\beta_6$ (Lane 6) | 0.0243 | 0.0258† | -0.0406 | -0.0411* | -0.0044 | -0.0043 | -0.0013 |
| | (0.0211) | (0.0190) | (0.0318) | (0.0246) | (0.0190) | (0.0156) | (0.0168) |
| $\beta_7$ (Lane 7) | 0.0216 | 0.0166 | -0.0442† | -0.0353† | -0.0143 | -0.0106 | -0.0125 |
| | (0.0232) | (0.0184) | (0.0293) | (0.0246) | (0.0190) | (0.0155) | (0.0166) |
| $\beta_8$ (Lane 8) | 0.0564** | 0.0375* | -0.0258 | -0.0143 | 0.0146 | 0.0142 | 0.0121 |
| | (0.0230) | (0.0195) | (0.0324) | (0.0278) | (0.0198) | (0.0171) | (0.0181) |
| $\beta_9$ (Lane 9) | -0.0039 | -0.0008 | -0.0686* | -0.090*** | -0.0431† | -0.0513** | -0.0594** |
| | (0.0355) | (0.00327) | (0.0398) | (0.0283) | (0.0280) | (0.0218) | (0.0239) |
| $\alpha_1$ (Wind) | -0.0446*** | -0.0429*** | -0.0555*** | -0.0532*** | -0.0508*** | -0.0490*** | -0.0481*** |
| | (0.0100) | (0.0058) | (0.0061) | (0.006) | (0.0056) | (0.0041) | (0.0045) |
| $\alpha_2$ (Reaction Time) | 1.01*** | 0.8025*** | 1.252*** | 1.220*** | 1.089*** | 0.992*** | 0.940*** |
| | (0.253) | (0.1802) | (0.3187) | (0.233) | (0.211) | (0.151) | (0.159) |
| $\alpha_3$ (PB) | 0.484*** | 0.448*** | 0.2563*** | 0.4095*** | 0.355*** | 0.444*** | 0.391*** |
| | (0.0688) | (0.0573) | (0.1026) | (0.0799) | (0.0779) | (0.0534) | (0.0562) |
| $\alpha_4$ (SB) | 0.351*** | 0.349*** | 0.6797*** | 0.4568*** | 0.549*** | 0.399*** | 0.429*** |
| | (0.0686) | (0.0583) | (0.1079) | (0.0861) | (0.0820) | (0.0565) | (0.0600) |
| $\alpha_5$ (Male) | | | | | -0.106*** | -0.178*** | -0.235*** |
| | | | | | (0.0187) | (0.0164) | (0.0201) |
| $\alpha_0$ (constant) | 1.731*** | 2.152*** | 0.7524*** | 1.557*** | 1.113*** | 2.151*** | 0.4742*** |
| | (0.278) | (0.211) | (0.222) | (0.216) | (0.184) | (0.156) | (0.179) |
| N | 828 | 786 | 788 | 748 | 1616 | 1534 | 1285 |
| $R^2$ | 0.684 | 0.759 | 0.888 | 0.854 | 0.920 | 0.946 | 0.952 |
| Outliers Removed | No | Yes | No | Yes | No | Yes | Yes |
| Pos 1/2 Removed | No | No | No | No | No | No | Yes |

This table reports the coefficients estimated from model Eq (4). To ease interpretation of the results, the independent variable associated with each coefficient estimate is highlighted in parentheses. Robust standard errors are reported in parentheses.

†, *, **, and *** denote significance at the one-sided 10%, two-sided 10%, 5%, and 1% levels, respectively.

be interested in preserving energy for later races and, as such, give "just enough" effort to advance. The concern is that these "just enough" effort types may supply different levels of effort conditional on their lane assignments, which may impact the estimates of lane effects. It is typical that two or three racers qualify to advance. The IAAF rules that determine qualification for later rounds vary by meet as they can be determined by Technical Delegates. However, it is common that two or three racers from each heat automatically advance, with the possibility of more runners qualifying on time. Aside from these "just enough" types, the remaining athletes are likely to be "maximum effort" types in attempting to qualify to advance. While it is certainly plausible that differential effort provision conditional on lanes could exist, it is

important to note that these athletes would constitute the minority of runners because of qualification rules. As an additional robustness check, I re-estimate the model Eq (4) excluding runners who finished in first or second place. Because excluding two runners per race amounts to an important reduction in sample size, I do this on the pooled data. Excluding these runners does not have a meaningful impact on the results, reported in the final column of Table 2, and suggests that differential provision of effort across lanes does not impact the estimates of lane effects. Collectively, these results suggest that there is no robust evidence of lane effects in 100m races. To ease interpretation of the results, Fig 1a plots the lane coefficient estimates from the second last column of Table 2 (i.e. the results generated from pooled data excluding outliers).

An issue that is relevant throughout this paper is statistical power. From a null finding of lane effects one cannot, of course, conclude that no lane effects exist. One can only conclude that given the statistical power in this analysis, if true lane effects do exist, their magnitude was not detectable. To provide a sense of the role that statistical power is playing in these null results I briefly highlight the lane effects that could be detected given this sample size. Following [20] I report some Minimum Detectable Effects (MDE) from the above regressions. Analyzing MDEs is a common approach to evaluate ex-post statistical power (see, e.g., [21]). At statistical power of 0.8 and a significance level of 5% or 10%, the MDE is found by multiplying the standard error on the coefficient estimate by 2.8 and 2.49, respectively. For example, using the results from the pooled data in column 6 in Table 2, the standard error on the lane 8 coefficient is 0.0171. Thus, the MDEs at the 5 or 10% significance level would be 0.0479 and 0.0426, respectively. While one cannot rule out true lane effects in the 100m from the null results in Table 2, these MDEs help establish that if lane effects do exist in the 100m, they must be quite small.

## 200m

200m races are more generally thought to have lane advantages and a common view is that periphery lanes—outside and inside lanes—are slower. Here I repeat the same general analysis strategy as above. To begin, the randomization check is reported in Table 9 in the S1 Appendix. These results show the randomization is effective. Only in lane 1 of the pooled data does season's best appear to be (weakly) related to lane assignments, which again could be a result of many fewer observations in lane 1.

Results from running model Eq (4) on the 200m data are reported in Table 3. The Men's, Women's, and Pooled data including or excluding outliers all show evidence that outside lanes produce lower average race times than lane 2. This consistency across Men's and Women's races, and the fact that these lane advantages seem to monotonically increase as the lane number increases are reassuring results. The estimated lane coefficients using pooled data and excluding outliers are plotted in Fig 1b. The estimates are also robust to excluding runners who finish first or second in each race, reported in the final column of Table 3. As discussed above, this suggests that differential provision of effort across lanes from faster athletes is not driving the results.

These estimates suggest the advantage of outside lanes can be sizable. For example, in the Women's data, excluding outliers, lane 8 is estimated to be 0.1781 faster than lane 2. The standard deviation (SD) of race times in this data is 0.68. A common way to estimate the magnitude of an effect is to compute the effect size $= \frac{|Effect|}{SD}$. Thus, these estimated results produce an effect size of 0.262, which is sizable. Put a different way, these lane effects could easily be the difference between qualifying, or not, to advance to the next round of the race.

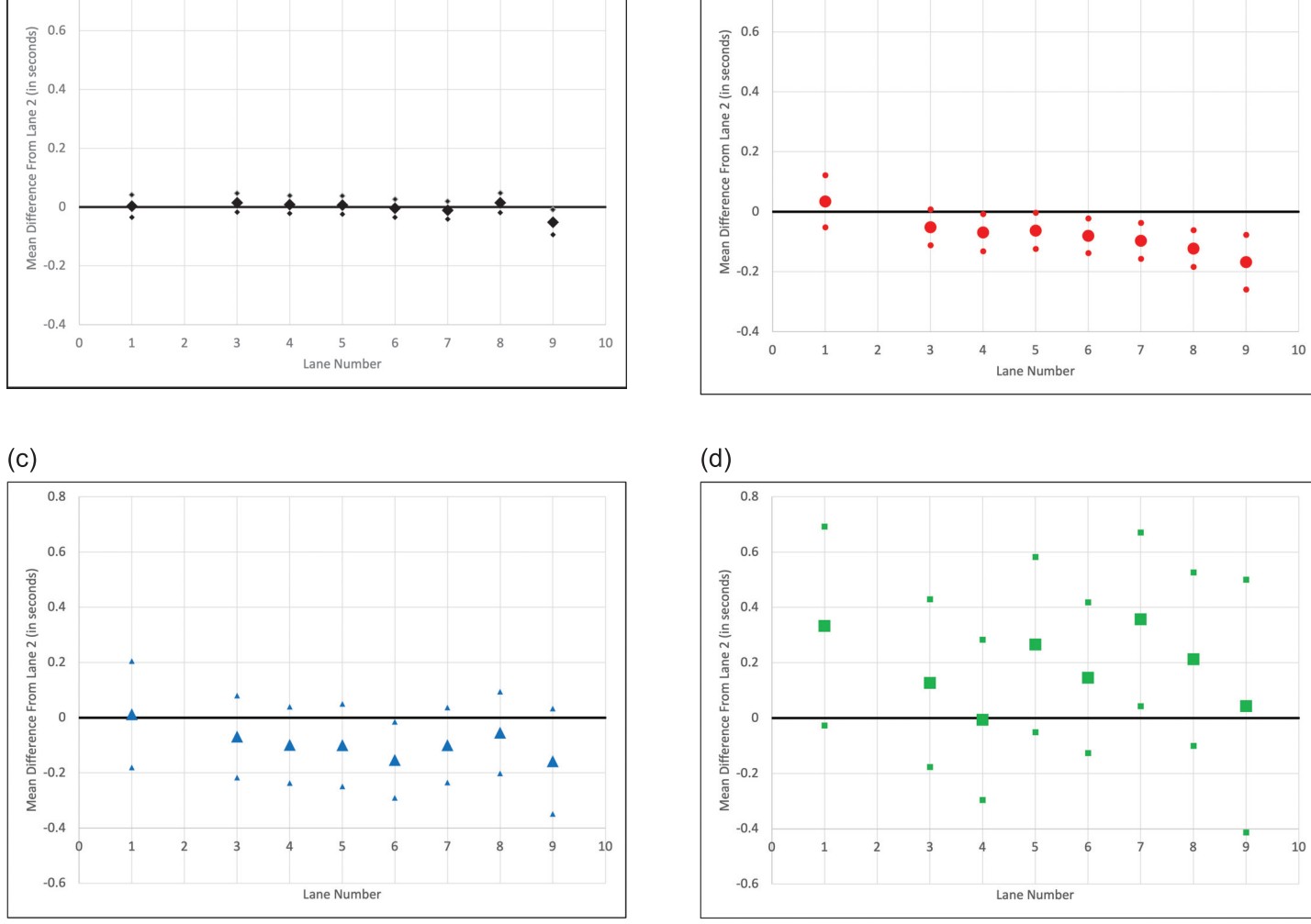

**Fig 1. Graphical display of regression results.** These figures plot the estimated lane effects using pooled men's and women's data and excluding outliers. 95% confidence intervals are denoted by the smaller symbols.

Of particular interest is the fact that these estimated lane advantages are the opposite of what is commonly believed regarding outside lanes. The seeming persistence and pervasiveness of false beliefs is interesting and I return to it in the Discussion section.

## 400m

Turning to the 400m races, I again first present the randomization check in Table 10, reported in the S1 Appendix. These results again show robust evidence that the randomization successfully balances racers by their season's bests across the different lanes. The only, weakly, significant result is for lane 3 in the Women's data, and this disappears when outliers are excluded.

The estimates of lane effects on race times in the 400m are somewhat consistent with the 200m, but are much noisier. These results are reported in Table 4. Wind speed is not recorded in the 400m, so these results are estimated by running model Eq (4) without wind as a control. There is some mixed evidence in the Women's data that outside lanes produce faster race

**Table 3. Regression results for 200m.**

| Coeff. (Ind. Var.) | Mens | | Womens | | Pooled | | |
|---|---|---|---|---|---|---|---|
| $\beta_1$ (Lane 1) | 0.0307 | 0.0856* | -0.0798 | -0.0524 | 0.0010 | 0.0351 | 0.0459 |
| | (0.0807) | (0.0448) | (0.0752) | (0.0774) | (0.0564) | (0.0441) | (0.0461) |
| $\beta_3$ (Lane 3) | -0.0295 | -0.0289 | -0.0645 | -0.0872† | -0.0421 | -0.0517* | -0.0645** |
| | (0.0730) | (0.0352) | (0.0561) | (0.0529) | (0.0481) | (0.0306) | (0.0312) |
| $\beta_4$ (Lane 4) | -0.0626 | -0.0540† | -0.0900† | -0.0966* | -0.0717† | -0.0691** | -0.0704** |
| | (0.0750) | (0.0350) | (0.0610) | (0.0584) | (0.0503) | (0.0317) | (0.0340) |
| $\beta_5$ (Lane 5) | -0.0930† | -0.0413 | -0.0677 | -0.0975* | -0.0812* | -0.0628** | -0.0653** |
| | (0.0692) | (0.0347) | (0.0617) | (0.0548) | (0.0481) | (0.0308) | (0.0323) |
| $\beta_6$ (Lane 6) | -0.0598 | -0.0343 | -0.1389*** | -0.1409*** | -0.0941* | -0.0796*** | -0.0842*** |
| | (0.0784) | (0.0345) | (0.0533) | (0.0524) | (0.0500) | (0.0294) | (0.0303) |
| $\beta_7$ (Lane 7) | -0.1035† | -0.0595* | -0.1598*** | -0.1472*** | -0.1263*** | -0.0967*** | -0.0804** |
| | (0.0714) | (0.0349) | (0.0544) | (0.0533) | (0.0467) | (0.0306) | (0.0321) |
| $\beta_8$ (Lane 8) | -0.1270* | -0.0838** | -0.1621*** | -0.1781*** | -0.1396*** | -0.1222*** | -0.1059*** |
| | (0.0706) | (0.0350) | (0.0610) | (0.0550) | (0.0484) | (0.0313) | (0.0329) |
| $\beta_9$ (Lane 9) | -0.1871** | -0.114** | -0.232*** | -0.2417*** | -0.2040*** | -0.1677*** | -0.1508*** |
| | (0.0860) | (0.0523) | (0.0795) | (0.0802) | (0.0586) | (0.0465) | (0.0498) |
| $\alpha_1$ (Wind) | -0.0623*** | -0.0608*** | -0.1023*** | -0.0853*** | -0.0809*** | -0.0719*** | -0.0660*** |
| | (0.0164) | (0.0087) | (0.0147) | (0.0142) | (0.0112) | (0.0081) | (0.0086) |
| $\alpha_2$ (Reaction Time) | 1.017** | 1.541*** | 0.9392** | 0.949** | 1.001*** | 1.325*** | 1.530*** |
| | (0.432) | (0.2789) | (0.4312) | (0.419) | (0.312) | (0.2449) | (0.2518) |
| $\alpha_3$ (PB) | 0.463*** | 0.385*** | 0.5875*** | 0.4966*** | 0.5464*** | 0.4491*** | 0.4229*** |
| | (0.0673) | (0.0507) | (0.0654) | (0.0580) | (0.0509) | (0.0407) | (0.0430) |
| $\alpha_4$ (SB) | 0.469*** | 0.475*** | 0.2832*** | 0.350*** | 0.346*** | 0.4027*** | 0.4162*** |
| | (0.0739) | (0.0548) | (0.0710) | (0.0632) | (0.0584) | (0.0447) | (0.0480) |
| $\alpha_5$ (Male) | | | | | -0.312*** | -0.432*** | -0.515*** |
| | | | | | (0.0643) | (0.0396) | (0.0501) |
| $\alpha_0$ (constant) | 1.674*** | 3.013*** | 3.383*** | 3.918*** | 2.823*** | 3.677*** | 3.985*** |
| | (0.5837) | (0.3728) | (0.716) | (0.461) | (0.553) | (0.324) | (0.382) |
| N | 926 | 880 | 708 | 671 | 1634 | 1551 | 1335 |
| $R^2$ | 0.647 | 0.75 | 0.81 | 0.757 | 0.931 | 0.958 | 0.960 |
| Outliers Removed | No | Yes | No | Yes | No | Yes | Yes |
| Pos 1/2 Removed | No | No | No | No | No | No | Yes |

This table reports the coefficients estimated from model (4). To ease interpretation of the results, the independent variable associated with each coefficient estimate is highlighted in parentheses. Robust standard errors are reported in parentheses.

†, *, **, and *** denote significance at the one-sided 10%, two-sided 10%, 5%, and 1% levels, respectively.

times, consistent with the lane advantages estimated in the 200m races. However, they do not appear to be monotonically decreasing with lane number, and, in addition, they are absent in the Men's data. When the data is pooled together and outliers are excluded lanes 4, 5, 6, 7 and 9 show some evidence of faster race times relative to lane 2. These results are not greatly impacted by excluding runners who finish in first or second (reported in the final column). The estimates using pooled data and excluding outliers are graphically depicted in Fig 1c. Visually, the results between the 200 and 400m look somewhat similar, with race times tending to increase with lane number, but from the 95% confidence intervals, it is clear that the 400m results are statistically weaker.

**Table 4. Regression results for 400m.**

| Coeff. (Ind. Var.) | Mens | | Womens | | Pooled | | |
|---|---|---|---|---|---|---|---|
| $\beta_1$ (Lane 1) | 0.0531 | 0.0526 | 0.0717 | -0.0357 | 0.0597 | 0.0123 | -0.0234 |
| | (0.1126) | (0.1035) | (0.2403) | (0.1921) | (0.1160) | (0.0981) | (0.0993) |
| $\beta_3$ (Lane 3) | -0.0123 | -0.0251 | 0.1369 | -0.1217 | 0.0431 | -0.0689 | -0.0358 |
| | (0.0929) | (0.0780) | (0.1726) | (0.1472) | (0.0906) | (0.0758) | (0.0805) |
| $\beta_4$ (Lane 4) | -0.0536 | -0.0302 | -0.1042 | -0.1875† | -0.0760 | -0.0987† | -0.0952 |
| | (0.0908) | (0.0839) | (0.1343) | (0.1229) | (0.0770) | (0.0704) | (0.0760) |
| $\beta_5$ (Lane 5) | -0.0900 | -0.1009 | -0.0273 | -0.0990 | -0.0608 | -0.0994† | -0.0997 |
| | (0.1055) | (0.0882) | (0.1439) | (0.1348) | (0.0857) | (0.0761) | (0.0827) |
| $\beta_6$ (Lane 6) | 0.0454 | -0.0551 | -0.2576** | -0.2674** | -0.0953 | -0.1528** | -0.1296* |
| | (0.1228) | (0.0849) | (0.1222) | (0.1219) | (0.0862) | (0.0703) | (0.0735) |
| $\beta_7$ (Lane 7) | -0.0005 | -0.0350 | -0.1512 | -0.1744† | -0.0703 | -0.0991† | -0.0942 |
| | (0.1019) | (0.0839) | (0.1228) | (0.1196) | (0.0778) | (0.0693) | (0.0738) |
| $\beta_8$ (Lane 8) | -0.0525 | -0.0535 | -0.0775 | -0.0639 | -0.0634 | -0.0540 | -0.0290 |
| | (0.0975) | (0.0860) | (0.1376) | (0.1343) | (0.0809) | (0.0754) | (0.0798) |
| $\beta_9$ (Lane 9) | 0.0178 | 0.0150 | -0.3458** | -0.3465** | -0.1547† | -0.1578† | -0.1527† |
| | (0.1234) | (0.1138) | (0.1578) | (0.1607) | (0.0996) | (0.0975) | (0.0997) |
| $\alpha_1$ (Reaction Time) | 1.272** | 1.493*** | 2.319*** | 1.781*** | 1.807*** | 2.070*** | 1.927*** |
| | (0.600) | (0.442) | (0.741) | (0.6689) | (0.4697) | (0.385) | (0.412) |
| $\alpha_2$ (PB) | 0.6397*** | 0.5588*** | 0.700*** | 0.5851*** | 0.6760*** | 0.5750*** | 0.5834*** |
| | (0.0699) | (0.0622) | (0.0845) | (0.0755) | (0.0572) | (0.0523) | (0.0577) |
| $\alpha_3$ (SB) | 0.3667*** | 0.3344*** | 0.2691*** | 0.2958*** | 0.3099*** | 0.3100*** | 0.3050*** |
| | (0.0746) | (0.0667) | (0.0883) | (0.0826) | (0.0604) | (0.0573) | (0.0638) |
| $\alpha_4$ (Male) | | | | | -0.3683*** | -0.9782*** | -1.015*** |
| | | | | | (0.1329) | (0.1034) | (0.1356) |
| $\alpha_0$ (constant) | 0.2126 | 5.337*** | 2.250*** | 6.784*** | 1.467† | 6.629*** | 6.532*** |
| | (1.359) | (0.9447) | (1.330) | (1.107) | (1.025) | (0.778) | (0.951) |
| N | 872 | 826 | 677 | 643 | 1549 | 1469 | 1269 |
| $R^2$ | 0.793 | 0.748 | 0.79 | 0.758 | 0.947 | 0.9605 | 0.9610 |
| Outliers Removed | No | Yes | No | Yes | No | Yes | Yes |
| Pos 1/2 Removed | No | No | No | No | No | No | Yes |

This table reports the coefficients estimated from model (4). To ease interpretation of the results, the independent variable associated with each coefficient estimate is highlighted in parentheses. Robust standard errors are reported in parentheses.

†, *, **, and *** denote significance at the one-sided 10%, two-sided 10%, 5%, and 1% levels, respectively.

One important issue with the 400m, and 800m below, is that the longer average race times tend to be associated with greater dispersion in race times. For example, as noted above, the standard deviation of race times excluding outliers in the 200m Women's data is 0.68. The analogous standard deviation in the 400m is 1.64 seconds. As a result, for a given number of observations, statistical power weakens as event times increase. To give a sense of statistical power, I again report the MDE for the 400m. For example, using the pooled data without outliers, the estimate for lane 8 has a standard error of 0.0754. With statistical power of 0.8, this gives a MDE of 0.2111 and 0.1878 for the 5% and 10% significance levels, respectively. Thus, given the number of observations in the 400m data, statistical power would be insufficient to pick up lane effects that would be similar in magnitude as the 200m. Of course, it is also important the emphasize that even if there are lane effects in the 400m that are of similar magnitude

as the 200m, their relative importance would be much smaller in the 400m since they represent a much smaller fraction of the mean or standard deviation of race times.

Also of interest is that these results are in contrast to the common belief that outside lanes are a significant disadvantage in the 400m. In both [1, 2] there is discussion about the gold medal race in the 2016 Olympics by Wayde van Niekerk. He is the first man to win the 400m from lane 8 and these articles clearly highlight the sentiment that this is impressive because lane 8 places runners at a disadvantage. However, the results in Table 4 show that, if anything, outside lanes produce average race times that are *faster* than lane 2. Of course, winning from lane 8 is impressive in that the runner registered one of the slowest qualifying times for the final, but this does not necessarily suggest that lane 8 itself is a disadvantage: van Niekerk's improvement from his semifinal time was an impressive 1.42 seconds, where as the average improvement of all the other runners in that race was 0.172 seconds.

### 800m

The 800m race is unique from the above events as lane assignments are not fixed for the duration of the race. Runners are assigned to a lane and must remain in that lane until the break line 100m from the start. This unique feature of the 800m, relative to the other shorter distance events, makes it interesting to explore in the context of lane assignment effects.

I again begin with the randomization check for the 800m data, reported in Table 11 in the S1 Appendix. There appears to be robust evidence that the randomization is effective. Moving on to the estimates of lane effects, I again implement model Eq (4). However, wind speed and reaction times are not recorded for the 800m and are thus not included in the regression. In addition, since the 800m tends to be a pack race—runners tend to run together in a pack for some portion of the race—I also include race fixed effects. Thus, lane effects are estimated after controlling for the average time in a race. On occasion, when tracks do not have a ninth lane, 800m races can have two runners assigned to lane 8. This is quite rare in the data, but I exclude these racers when it does occurs. These regression results are reported in Table 5.

The results are somewhat mixed, possibly due to the issues regarding longer race times and dispersion highlighted above, but there is some weak evidence that outside lanes tend to produce slower race times on average. For example, in the pooled data excluding outliers, lanes 5, 7 and 8 show positive and significant (weakly in some cases) effects on race times, ranging from 0.213 to 0.357 seconds. These results are generally consistent, but somewhat statistically weaker, when runners who finish in first or second are excluded. The results using pooled data and excluding outliers are reported in Fig 1d.

Of interest, the result that outside lanes produce slower race times on average is the opposite of the general result found in the 200m and 400m. As noted above, one possible explanation for this may be the unique lane break feature of the 800m. Since the inside lane of the track minimizes the distance covered, after the break-line all runners converge to the inside lanes. This might make the inside lanes advantageous as runners in the outside lanes either have to jockey for position with runners who have an establish position on the inside of the track, or continue to run in lanes which lengthen the distance travelled around the track.

### Vantage points and effort effects

As noted above, the narrative that outside lanes are undesirable in races with corners stems from the idea that not being able to see competitors puts runners at a disadvantage. It could be the case that seeing a competitor generates additional motivation for runners and spurs increased effort. Because of staggered starts, higher lane numbers will be able to see fewer runners, and the outermost lane can see no other runners (until they are passed). This effect will

**Table 5. Regression results for 800m.**

| Coeff. (Ind. Var.) | Mens | | Womens | | Pooled | | |
|---|---|---|---|---|---|---|---|
| $\beta_1$ (Lane 1) | 0.2290 | 0.1579 | 0.6594 | 0.5756** | 0.3135 | 0.333* | 0.2244 |
| | (0.4478) | (0.222) | (0.6427) | (0.2872) | (0.3725) | (0.1835) | (0.1900) |
| $\beta_3$ (Lane 3) | 0.0351 | 0.1168 | -0.0693 | 0.1869 | -0.1469 | 0.1270 | 0.1981 |
| | (0.3065) | (0.1929) | (0.5944) | (0.2327) | (0.2969) | (0.1547) | (0.1592) |
| $\beta_4$ (Lane 4) | 0.6195 | -0.0455 | -0.1565 | 0.1541 | 0.1903 | -0.0061 | 0.0879 |
| | (0.6913) | (0.1656) | (0.5726) | (0.2264) | (0.4413) | (0.1462) | (0.1476) |
| $\beta_5$ (Lane 5) | -0.0641 | -0.1209 | 0.4735 | 0.7145** | 0.1108 | 0.2660* | 0.2474† |
| | (0.2896) | (0.1742) | (0.5744) | (0.2923) | (0.2796) | (0.1616) | (0.1722) |
| $\beta_6$ (Lane 6) | -0.0958 | 0.0446 | 0.3534 | 0.1648 | 0.0583 | 0.1461 | 0.1446 |
| | (0.3152) | (0.1690) | (0.6474) | (0.2644) | (0.3148) | (0.1389) | (0.1510) |
| $\beta_7$ (Lane 7) | 0.4814† | 0.2301 | 0.1087 | 0.5689** | 0.2314 | 0.3570** | 0.3834** |
| | (0.3506) | (0.1913) | (0.5692) | (0.2679) | (0.2991) | (0.1602) | (0.1771) |
| $\beta_8$ (Lane 8) | 0.0114 | -0.0435 | 0.9580† | 0.5302* | 0.3375 | 0.2131† | 0.1522 |
| | (0.3157) | (0.1832) | (0.6791) | (0.2764) | (0.3211) | (0.1600) | (0.1709) |
| $\beta_9$ (Lane 9) | 2.029 | 0.1053 | -0.7562 | 0.0782 | 0.6528 | 0.0437 | 0.1797 |
| | (1.823) | (0.2424) | (0.8611) | (0.3968) | (1.085) | (0.2330) | (0.2198) |
| $\alpha_1$ (PB) | 0.7129*** | 0.2647*** | 0.4550** | 0.1629† | 0.531** | 0.218* | 0.490*** |
| | (0.1879) | (0.0691) | (0.2158) | (0.1182) | (0.2228) | (0.1257) | (0.0312) |
| $\alpha_2$ (SB) | 0.331*** | 0.280*** | 0.2558† | 0.1038* | 0.2923* | 0.1364* | 0.0410** |
| | (0.0904) | (0.0731) | (0.1558) | (0.0570) | (0.1746) | (0.0755) | (0.0162) |
| $\alpha_3$ (Male) | | | | | -5.47** | -16.43*** | -12.65*** |
| | | | | | (2.717) | (1.918) | (1.051) |
| $\alpha_0$ (constant) | -2.016 | 49.61*** | 36.88** | 90.11*** | 27.31** | 86.04*** | 63.55*** |
| | (16.12) | (3.13) | (14.56) | (12.00) | (12.37) | (9.15) | (3.74) |
| N | 803 | 762 | 613 | 582 | 1416 | 1344 | 1169 |
| $R^2$ | 0.565 | 0.816 | 0.700 | 0.817 | 0.886 | 0.976 | 0.9795 |
| Outliers Removed | No | Yes | No | Yes | No | Yes | Yes |
| Pos 1/2 Removed | No | No | No | No | No | No | Yes |

This table reports the coefficients estimated from model (4). To ease interpretation of the results, the independent variable associated with each coefficient estimate is highlighted in parentheses. Robust standard errors are reported in parentheses.

†, *, **, and *** denote significance at the one-sided 10%, two-sided 10%, 5%, and 1% levels, respectively.

likely be the most dramatic in the 200 and 400m. If these "effort effects" of lanes do exist, a natural interpretation is that they would cause average race times to increase with lane number (i.e. outside lanes would be slower). This effect goes in the opposite direction compared to the biomechanical effects of tight corners. The results reported above should be thought of as the *net* effects of lane assignments. It is possible that both margins (effort and biomechanical effects) impact runners, but the results in the 200 and 400m suggest that, if anything, race times decrease with lane number. As such, the narrative that outside lanes are undesirable because of effort effects is not well supported by the data.

This, of course, does not rule out that effort effects are active, just that the tight corner effects dominate. While the net effects are clearly what ultimately matters in terms of assessing the desirability of lanes, it is still an interesting question to know if effort effects are present. To assess this question, I revisit the 200 and 400m results and leverage the fact that it is not always the same lane which is the outermost one. Because not all lanes are full in each race and/or

some tracks do not have a 9th lane, there is some variability in which lane is the outermost (commonly lanes 8 or 9, and occasionally 7). Because the outermost lane and lane numbers are not perfectly correlated, I can leverage this fact to estimate the separate effect of being in the outermost lane (i.e. not having any competitors ahead of you to start a race).

To estimate these effects I run the same model as Eq (4) but where I also include an additional indicator variable taking a value of 1 (0) when a runner is (is not) in the outermost lane. In other words, controlling for lane effects as in Eq (4), is there a separate statistical effect of being in the outermost lane? To cut down on repetition, these results are estimated on the data without outliers and are reported in Table 6.

**Table 6. Regression results for 200m and 400m with a separate indicator variable for the outermost lane.**

| Coeff. (Ind. Var.) | 200m | | | 400m | | |
|---|---|---|---|---|---|---|
| | **Mens** | **Womens** | **Pooled** | **Mens** | **Womens** | **Pooled** |
| $\beta_1$ (Lane 1) | 0.0856† | -0.0524 | 0.0351 | 0.0525 | -0.0347 | 0.0124 |
| | (0.0524) | (0.0774) | (0.0441) | (0.104) | (0.1922) | (0.0981) |
| $\beta_3$ (Lane 3) | -0.0285 | -0.0873† | -0.0516* | -0.0251 | -0.1210 | -0.0689 |
| | (0.0332) | (0.0530) | (0.0296) | (0.0800) | (0.1474) | (0.0758) |
| $\beta_4$ (Lane 4) | -0.0536† | -0.0971* | -0.0691** | -0.0302 | -0.1871† | -0.0988† |
| | (0.0333) | (0.0585) | (0.0317) | (0.0839) | (0.1231) | (0.0704) |
| $\beta_5$ (Lane 5) | -0.0407 | -0.0977* | -0.0628** | -0.1009 | -0.0985 | -0.0994† |
| | (0.0344) | (0.0548) | (0.0308) | (0.0883) | (0.1350) | (0.0761) |
| $\beta_6$ (Lane 6) | -0.0341 | -0.1409*** | -0.0796*** | -0.0551 | -0.2669** | -0.1529** |
| | (0.0334) | (0.0525) | (0.0294) | (0.0849) | (0.1220) | (0.0703) |
| $\beta_7$ (Lane 7) | -0.0608* | -0.1499*** | -0.0989*** | -0.0336 | -0.1846† | -0.1010† |
| | (0.0353) | (0.0533) | (0.0306) | (0.0841) | (0.1189) | (0.0692) |
| $\beta_8$ (Lane 8) | -0.1526*** | -0.2182*** | -0.1761*** | 0.0002 | -0.1948 | -0.0910 |
| | (0.0479) | (0.0689) | (0.0405) | (0.1418) | (0.1545) | (0.1035) |
| $\beta_9$ (Lane 9) | -0.2110*** | -0.3020*** | -0.2458*** | 0.0948 | -0.5781** | -0.2171† |
| | (0.0748) | (0.1063) | (0.0623) | (0.1807) | (0.2441) | (0.1498) |
| $\alpha_1$ (Outermost) | 0.0973* | 0.0597 | 0.0778* | -0.0798 | 0.2315 | 0.0591 |
| | (0.0499) | (0.0682) | (0.0410) | (0.1400) | (0.1829) | (0.1128) |
| $\alpha_2$ (Wind) | -0.0610*** | -0.0861*** | -0.0726*** | | | |
| | (0.009) | (0.0143) | (0.0081) | | | |
| $\alpha_3$ (Reaction Time) | 1.498*** | 0.954** | 1.309*** | 1.502*** | 2.694*** | 2.073*** |
| | (0.290) | (0.416) | (0.2443) | (0.443) | (0.671) | (0.386) |
| $\alpha_4$ (PB) | 0.387*** | 0.501*** | 0.453*** | 0.559*** | 0.5829*** | 0.5746*** |
| | (0.0573) | (0.0581) | (0.0407) | (0.0623) | (0.0755) | (0.0523) |
| $\alpha_5$ (SB) | 0.474*** | 0.346*** | 0.399*** | 0.334*** | 0.2960*** | 0.3101*** |
| | (0.0640) | (0.0633) | (0.0448) | (0.0669) | (0.0824) | (0.0573) |
| $\alpha_6$ (Male) | | | -0.431*** | | | -0.981*** |
| | | | (0.0395) | | | (0.1037) |
| $\alpha_0$ (constant) | 3.00*** | 3.916*** | 3.671*** | 5.333*** | 6.884*** | 6.646*** |
| | (0.395) | (0.461) | (0.324) | (0.946) | (1.072) | (0.779) |
| N | 880 | 671 | 1551 | 826 | 643 | 1469 |
| $R^2$ | 0.752 | 0.757 | 0.958 | 0.615 | 0.759 | 0.9605 |
| Outliers Removed | Yes | Yes | Yes | Yes | Yes | Yes |

This table reports the coefficients estimated from model (4) with a separate indicator variable for the outermost lane. To ease interpretation of the results, the independent variable associated with each coefficient estimate is highlighted in parentheses. Robust standard errors are reported in parentheses.

†, *, **, and *** denote significance at the one-sided 10%, two-sided 10%, 5%, and 1% levels, respectively.

The coefficient of interest is on the outermost indicator variable ($\alpha_1$). In the men's 200m, the estimated coefficient is 0.0973 with a p-value of 0.052. So there is reasonable statistical evidence that, after controlling for lane effects, being in the outermost lane does generate slower race times on average. In the women's 200m, the coefficient falls to 0.0597 and is insignificant at standard levels. And in the pooled data the coefficient is 0.0778 with a p-value of 0.058. Overall, while the evidence is somewhat mixed between the men's and women's races, there is some evidence that being in the outmost lane does have a negative impact on runners. Again, it is important to emphasize that these results do not mean that the outside lanes generate an *overall* slowdown in race times. The results above clearly highlight that being in the outside lanes in the 200m generate, on average, faster race times. The positive coefficients on the outermost variable is simply the marginal impact of being in the outermost lane.

The results from the 400m races are much more mixed, with none of the outermost coefficients being statistically significant. Again, this could stem from weaker statistical power at this distance.

## Alternative regression models

A desirable feature of model Eq (4) is that it is agnostic about the structure of lane advantages, allowing each lane to have a separate treatment effect. However, one of the downsides about this approach is that it requires eight regressors, which compromises statistical power. This may be especially worrisome in the 400 and 800m where statistical power issues are more salient. In this section I explore two alternative statistical models to help to alleviate the statistical power issue. In the first approach, I repeat the general approach in Eq (4) but instead pool runners together in lanes 1 and 2, 3 and 4, 5 and 6, and 7, 8, and 9. This dramatically increases the number of observations per regressor but, of course, has the obvious downside of assuming the statistical impact of, for example, lanes 3 and 4 is identical. Specifically, with this alternative regression I estimate:

$$Y_{i,j} = \alpha_0 + \beta_1 T_{i,j}^{3,4} + \beta_2 T_{i,j}^{5,6} + \beta_3 T_{i,j}^{7,8,9} + \alpha_f X_{f,i,j} + \epsilon_{i,j} \qquad (6)$$

Where $T^{3,4}$, $T^{5,6}$, and $T^{7,8,9}$ denote dummy variables which take a value of 1 when racers are in lanes 3 or 4, 5 or 6, and 7 or 8 or 9, respectively, and take a value of 0 otherwise. Lanes 1 and 2 are now the baseline grouping. To reduce repetition, I report only results which pool men's and women's data and exclude outliers.

Results from Eq (6) are reported in Table 7. Overall, the general results are quite similar to those generated from the original model Eq (4). There is no evidence of lanes assignments impacting races times in the 100m and there is strong evidence that outsides produce faster race times in the 200m. The results from the 400m become somewhat more significant and show some evidence that outside lanes in the 400m are also faster. And in the 800m, outside lanes tend to produce slower race times, but the effects remain quite weak statistically.

For the last model, instead of utilizing indicator variables for lane assignments I implement a continuous lane variable. In particular, I estimate the following statistical model:

$$Y_{i,j} = \alpha_0 + \beta_1 Z_{i,j} + \alpha_f X_{f,i,j} + \epsilon_{i,j} \qquad (7)$$

where $Z_{i,j}$ is a continuous lane variable taking values from 1 to 9. Of course, the implied assumption here is that lane numbers impact race times in a linear fashion. This further increases statistical power, but, of course, has the undesirable feature of imposing a functional form which may or may not capture the true data generating process. Results of this regression are reported in Table 8.

**Table 7. Regression results for coarser lane groupings.**

| Coeff. (Ind. Var.) | 100m | 200m | 400m | 800m |
|---|---|---|---|---|
| $\beta_1$ (Lane 3, 4) | 0.0105 | -0.0700*** | -0.0880† | -0.0409 |
| | (0.0121) | (0.0244) | (0.0572) | (0.1242) |
| $\beta_2$ (Lane 5, 6) | 0.0002 | -0.0806*** | -0.1295** | 0.1047 |
| | (0.0120) | (0.0240) | (0.0573) | (0.1172) |
| $\beta_3$ (Lane 7, 8, 9) | -0.0064 | -0.1257*** | -0.0919* | 0.1570† |
| | (0.0121) | (0.0241) | (0.0549) | (0.1215) |
| $\alpha_1$ (Wind) | -0.0487*** | -0.0720*** | | |
| | (0.0041) | (0.0081) | | |
| $\alpha_2$ (Reaction Time) | 0.994*** | 1.343** | 2.068*** | |
| | (0.1507) | (0.244) | (0.384) | |
| $\alpha_3$ (PB) | 0.447*** | 0.447*** | 0.576*** | 0.220* |
| | (0.0527) | (0.0403) | (0.0523) | (0.1258) |
| $\alpha_4$ (SB) | 0.397*** | 0.4048*** | 0.310*** | 0.1344* |
| | (0.0557) | (0.0443) | (0.0574) | (0.0757) |
| $\alpha_5$ (Male) | -0.176*** | -0.4312*** | -0.9738*** | -16.45*** |
| | (0.0164) | (0.0396) | (0.1032) | (1.926) |
| $\alpha_0$ (constant) | 1.829*** | 3.683*** | 6.602*** | 86.09*** |
| | (0.1560) | (0.3235) | (0.777) | (10.90) |
| N | 1534 | 1551 | 1469 | 1344 |
| $R^2$ | 0.946 | 0.9579 | 0.9604 | 0.9756 |
| Outliers Removed | Yes | Yes | Yes | Yes |

This table reports the coefficients estimated from model Eq (6), utilizing coarser lane groupings. To ease interpretation of the results, the independent variable associated with each coefficient estimate is highlighted in parentheses. Robust standard errors are reported in parentheses.

†, *, **, and *** denote significance at the one-sided 10%, two-sided 10%, 5%, and 1% levels, respectively.

The results in Table 8 are again quite similar to those found from the original model Eq (4). The $\beta_1$ coefficient is small and highly insignificant in the 100m and it is negative and highly significant in the 200m. While the coefficient is negative in the 400m, and a similar magnitude as the 200m, it is only weakly significant (p-value = 0.109). Finally, the $\beta_1$ coefficient is positive in the 800m, but also fails significance at standard levels (p-value = 0.22).

In summary, employing these alternative statistical models seems to buttress the initial estimates of lane effects found from estimating model Eq (4). In all cases there is no evidence of lane effects in the 100m and robust evidence that outside lanes are faster in the 200m. With coarser lane groupings the evidence of faster outside lanes in the 400m becomes somewhat stronger, and this aligns with the effects seen in the 200m. And finally, through all the different statistical models the evidence from the 800m suggests outside lanes are slower, but these effects are quite weak statistically.

## Discussion

Leveraging a random assignment rule implemented in the first round of IAAF events, this paper provides causal estimates of lane assignments in sprint distance track and field events. I find no evidence of lane advantages in the 100m, which suggests that a runner's vantage point is inconsequential for their performance. In the 200m, I find robust evidence that outside lanes on the track produce faster race times. This result is consistent with the biomechanical evidence on the impact of tight corners on running speeds. While average race times in outside

**Table 8. Regression results with lane effects modeled linearly.**

| Coeff. (Ind. Var.) | 100m | 200m | 400m | 800m |
|---|---|---|---|---|
| $\beta_1$ (Lane) | -0.0021 | -0.0189*** | -0.0132† | 0.0214 |
| | (0.0018) | (0.00349) | (0.00822) | (0.0176) |
| $\alpha_1$ (Wind) | -0.0487*** | -0.0716*** | | |
| | (0.0041) | (0.00807) | | |
| $\alpha_2$ (Reaction Time) | 1.00*** | 1.319*** | 2.119*** | |
| | (0.1510) | (0.2445) | (0.381) | |
| $\alpha_3$ (PB) | 0.4453*** | 0.4475*** | 0.575*** | 0.221* |
| | (0.0524) | (0.0403) | (0.0525) | (0.1263) |
| $\alpha_4$ (SB) | 0.3980*** | 0.4054*** | 0.3106*** | 0.1344* |
| | (0.0554) | (0.0443) | (0.0576) | (0.0760) |
| $\alpha_5$ (Male) | -0.177*** | -0.4281*** | -0.9718*** | -16.44*** |
| | (0.0164) | (0.0395) | (0.1033) | (1.932) |
| $\alpha_0$ (constant) | 1.85*** | 3.680*** | 6.566*** | 86.00*** |
| | (0.1553) | (0.3225) | (0.7744) | (10.92) |
| N | 1534 | 1551 | 1469 | 1344 |
| $R^2$ | 0.946 | 0.9579 | 0.9603 | 0.9756 |
| Outliers Removed | Yes | Yes | Yes | Yes |

This table reports the coefficients estimated from model Eq (7) with lane effects modeled linearly. Robust standard errors are reported in parentheses.

†, *, **, and *** denote significance at the one-sided 10%, two-sided 10%, 5%, and 1% levels, respectively.

lanes are also faster in the 400m, the statistical evidence is somewhat weaker than the 200m. But it is important to note that statistical power becomes more of an issue in events with longer race times. Finally, I find some weak evidence that outside lanes in the 800m tend to produce slower race times, which may be a product of the unique lane break feature of the 800m.

There are a number of interesting points worth discussing. The first is the fact that results in the 200m and 400m suggest that the commonly held belief that middle lanes are best is incorrect. Why these seemingly false beliefs persist is an interesting question. One possible interpretation is that in most observations of track and field races, slower athletes are assigned to the periphery lanes of the track. For example, in the IAAF rules, after round 1, athletes are ranked by their round 1 race times and: "Three draws will be made: i) one for the four highest ranked athletes or teams to determine placings in lanes 3, 4, 5 and 6, ii) another for the fifth and sixth ranked athletes or teams to determine placings in lanes 7 and 8, and iii) another for the two lowest ranked athletes or teams to determine placings in lanes 1 and 2." Thus, the runners assigned to the periphery lanes are the slowest runners in the race. As another example, in the widely used track and field software called Hytek, used in Olympic trials and NCAA championships, the "standard lane preferences" option in the software ranks lanes from most preferred to least preferred as: 4, 5, 3, 6, 2, 7, 1, 8. Again, the slowest runners are assigned to the periphery lanes. Failure to account for this non-random assignment to lanes may reinforce the idea that periphery lanes are slower. While this can possibly explain the persistence of false beliefs about lane advantages it fails to explain the origin of the these lane assignment rules. One possible explanation of the origin of these rules is a technological constraint. I have heard, but unfortunately have not been able to find documentation to support, that assigning faster runners to the middle lanes was done in the hand timing era to make it easier for timers to see runners cross the finish line in an "inverted-V" pattern, with the middle runners crossing first. This would allow hand timers on either side of the track to observe runners in a sequential

finish, and make accurate timing easier. This is an interesting possible explanation: a technical constraint was the impetus for lane assignment rules, which themselves led to beliefs that runners in middle lanes perform the best because of a failure to account for non-random assignment to lanes. Beyond addressing the specific question of lane advantages, the results in this paper could also be viewed as an interesting case study in the persistence of false beliefs (e.g. [22, 23]).

These results are also interesting in the context of the design of lane assignment rules. Lane assignment rules are designed to "reward" the fastest qualifying times with advantageous lanes in later rounds. However, the results here suggest that estimated lane advantages are not consistent with the implied advantages in lane assignment rules. This opens an interesting discussion about fairness and whether these lane assignment rules should be modified. In addition, in track meets that use seed times to assign lanes in the first round of events, there is a question about the fairness of the competition. If the goal of competition is to put all athletes on an even playing field to begin the competition, events that use non-random lane assignments in the first round put some runners at an entrenched disadvantage.

There are a number of ways this work could be expanded. To begin, I have not examined lane advantages in sprint distance hurdle events or relays. Perhaps more interesting given the results from the 200m and 400m is to examine the effect of lane assignments for indoor events, which use a 200m track that has tighter corners than outdoor 400m tracks. It is also possible this methodology could be extended to examine the effect of lane assignments in other sports. If a similar random assignment rule is used at some point in the competition, one could examine this question in swimming, cycling, and speed skating events.

## Supporting information

**S1 File.**
(ZIP)

**S1 Appendix.**
(PDF)

## Acknowledgments

I thank Erick Gong for helpful discussions as well as the Editor, Roy Cerqueti, and three anonymous reviewers for helpful comments.

## Author Contributions

**Data curation:** David Munro.

**Formal analysis:** David Munro.

**Methodology:** David Munro.

**Writing – original draft:** David Munro.

**Writing – review & editing:** David Munro.

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
