## [Decision Letter · Decision Letter 0]

13 Jan 2022

PONE-D-21-35820

Are there lane advantages in track and field?

PLOS ONE

Dear Dr. Munro,

Thank you for submitting your manuscript to PLOS ONE. After careful consideration, we have decided that your manuscript does not meet our criteria for publication and must therefore be rejected.

I am sorry that we cannot be more positive on this occasion, but hope that you appreciate the reasons for this decision.

Yours sincerely,

Roy Cerqueti, Ph.D.

Academic Editor

PLOS ONE

Reviewers' comments:

Reviewer's Responses to Questions

**Comments to the Author**

1. Is the manuscript technically sound, and do the data support the conclusions?

Reviewer #1: Yes

Reviewer #2: No

2. Has the statistical analysis been performed appropriately and rigorously? 

Reviewer #1: Yes

Reviewer #2: No

3. Have the authors made all data underlying the findings in their manuscript fully available?

Reviewer #1: No

Reviewer #2: Yes

4. Is the manuscript presented in an intelligible fashion and written in standard English?

Reviewer #1: Yes

Reviewer #2: Yes

5. Review Comments to the Author

Reviewer #1: Overall, this paper is interesting and mostly well executed. However, it sometimes lacks in motivation and a deeper analysis that tries to establish the root causes for the (non-)findings, which could be done with the present data. There are also a few issues in the presentation, with the potential for improvement.

Motivation

- The main scientific relevance of the research question – as it is presented in the intro - currently rest with how much one trusts the claims in the cited literature on biomechanics and physical factors. The author should make the significance of this literature more clear if this is intended.

- If not, the author should make the other motivation for this paper – and the results’ potential wider implications - more clear.

- This applies in particular to the motivational/psychological factors at play.

Physics, psychics and measurement

- Given that not all 9 nine lanes are always filled and that each lane has its own average curvature (e.g., measured by radian covered per meter run and its concentration), could one not construct a related variable which measures and tests biomechanics argument directly (and continuously)?

- It should be made clear whether there are potentially opposing effect from other factors. For example Berger and Pope (2011) show that “being behind” spurs effort. People on inner tracks, if randomly assigned, are on average more often (literally) behind their opponents. Could this be an explanation for null effects? (see also Teeselink, van den Assem and van Dolder, 2021 for an opposing account)?

- By constructing appropriate measures (e.g., whether someone was on the outmost lane and could not see competitors vs accounting for lane curvature which might not co-vary perfectly if not all lanes are filled) one could account for – and test - different theories.

- I was kind of surprised that wind but not wind shadow created by opponents – where lane assignment is probably relevant – was discussed.

- What if the average skill athletes’ level in the first round of these tournaments is not high enough that the finer features of the biomechanics and other subtle factors affecting performance just don’t play out to be decisive? Maybe even the first round is highly challenging to get into and only achievable for pros, but without further contextual info that’s hard to judge.

Econometric specification and presentation

- Are 8 lane dummies really the best specification, especially when there are concerns about statistical power? If effects are expected to be monotone across lanes, why not a continuous (linear or quadratic) function?

- If, there is a good reason to not use a continuous specification, why can lanes not be grouped (e.g., into 1&2, 3&4, 5&6, 7&8) to achieve more power per coefficient?

- Reading essentially 5 times the same specification and table but with different data sets is kind of hard and makes it hard to compare findings. Why not only present the 100m table as an example in the main text and then depict the (relevant) line coefficient and their SE graphically (e.g., lanes on the x-axis, normalised coefficients on the y-axis with SE and a line connecting them). This graphical presentation could then be added for all other distance to the same graph (e.g. by stacking the connecting lines with the coefficients). All other tables and the randomness check can then go to an appendix and the paper would be much more comprehensive. (Ideally the graph would also depict a normalised interval for MDE for each line.)

Other (some minor, some not) issues

- “was put in place during in the 1985-86 rules under rule 141.11” reads like the rule were only in place in the years/season 1985/1986 while elsewhere we find “AAF World Championships and U20 World Championships from 2000 to 2019”. I suggest to clarify this (also footnote 3 which is hard to read).

- In general, I would advise to describe a bit more clearly how/when/how long the random assignment was introduced. Right now it reads just like it was and then a bunch of technical details. Maybe that can be presented more story-like.

- I would also advise to refer to measures in the flow of the text not by their variable names but by what they describe in order to improve readability.

- the variable SB is first described as “SB is the runner’s season best race time”, then as “ assigned lanes based on prior race results (proxied here by SB)”. Am I correct that the author tries to claim that results prior to race can be proxied by SB, e.g., the best result across the whole season?

- The sum in the regression equation should index over only 8 dummies, not 9, as one lane is the baseline.

- I did not find the data in the paper or an appendix, except for a link to a sports website. I would expect the authors to share, with the manuscript, i) the original dataset used, including any outliers or incomplete data dropped for the actual analysis, ii) a short description on how it was generated, iii) the script(s) used to analyse and pre-process the data and to generate all tables/figures.

Reviewer #2: The topic of the paper is undoubtedly interesting and appealing. However, I have very serious concerns on the validity of the results presented in the paper due to the very poor and inaccurate description of the applied statistical methodology. Following, the details of the review are divided into major and minor comments.

Major comments:

1) The general description of the statistical methodology applied in the paper is completely missing. This issue does not allow to appropriately evaluate the validity of the results reported in the paper. The author should to carefully describe the applied statistical methodology in details in a separate section, by reporting in a rigorous way the main theory (formulas, assumptions, and the corresponding references). Furthermore, the specifications of the statistical models in formulas (1) and (2) are completely inaccurate; for instance, subscripts are missing in these formulas, the random components are just reported as “error” rather than through the well-known statistical notation, models’ assumptions are completely missing.

2) My main concern relates to the validity of the results reported in the paper. Since the description of the statistical methodology is completely missing, from what I see, it seems that the author apply a linear regression model? At the same time, the author claims along the entire manuscript to estimate “the causal effect of line assignments on race times”. However, it is well-known that we cannot speak about a causal effect when considering the “classical” linear regression model. There are several specific approaches for causal inference, as for instance the potential outcomes framework, causal graphs and similar; however, nothing is mentioned in the paper on this point. The issue mentioned above is of crucial importance on the entire validity and interpretation of the results reported in this paper, and it should be carefully justified and explained in details.

3) Statistical model diagnostics are completely missing. They should be performed and reported in order to appropriately evaluate the estimated statistical models.

4) Tables no.1-no.8: in all the tables, the estimated coefficients are reported incorrectly as Lane 1, Lane 3, Wind, etc. For example, the author should to write β ^_1 rather than Lane 1, β ^_3 rather than Lane 3, α ^_1 rather than Wind, and so on. Moreover, R^2 are erroneously reported as “R ^ 2”, and furthermore they should be discussed.

Minor comments:

1) To the best of my knowledge, the PLOS One Guidelines for authors require that the references in the text are reported by numbers, and footnotes are not permitted. Please, correct.

6. PLOS authors have the option to publish the peer review history of their article (what does this mean?). If published, this will include your full peer review and any attached files.

Reviewer #1: No

Reviewer #2: No

- - - - -

---

## [Author Response · Author response to Decision Letter 0]

3 Feb 2022

Response to reviewer comments:

Reviewer 1:

Overall, this paper is interesting and mostly well executed. However, it sometimes lacks in motivation and a deeper analysis that tries to establish the root causes for the (non-)findings, which could be done with the present data. There are also a few issues in the presentation, with the potential for improvement.

Motivation

- The main scientific relevance of the research question – as it is presented in the intro - currently rest with how much one trusts the claims in the cited literature on biomechanics and physical factors. The author should make the significance of this literature more clear if this is intended.

- If not, the author should make the other motivation for this paper – and the results’ potential wider implications - more clear.

- This applies in particular to the motivational/psychological factors at play.

Response: Thanks for these suggestions. I have added more discussion in the introduction of the relevance of the paper for the literature examining the performance effects of motivational/psychological factors.

Physics, psychics and measurement

- Given that not all 9 nine lanes are always filled and that each lane has its own average curvature (e.g., measured by radian covered per meter run and its concentration), could one not construct a related variable which measures and tests biomechanics argument directly (and continuously)?

Response: This is interesting. From the biomechanics literature there is no single “mechanism” that would slow runners down in tighter corners. Some ideas are that tight corners increase step frequency, lower foot force production, create asymmetries between legs, etc. Furthermore, while one can model the geometry of the track mathematically, to my knowledge there are not mathematical models relating this geometry to biomechanical factors. Without such a theory it’s hard to know how to treat the effect of curvature. For example, is the curvature effect linear, non-linear, etc. What I like about the baseline regression specification is that it’s agnostic about the lane-specific treatment effects. In relation to one of your points below, I have added two more specifications to the paper to explore the robustness of the results to other specifications.

- It should be made clear whether there are potentially opposing effect from other factors. For example Berger and Pope (2011) show that “being behind” spurs effort. People on inner tracks, if randomly assigned, are on average more often (literally) behind their opponents. Could this be an explanation for null effects? (see also Teeselink, van den Assem and van Dolder, 2021 for an opposing account)?

Response: See next response.

- By constructing appropriate measures (e.g., whether someone was on the outmost lane and could not see competitors vs accounting for lane curvature which might not co-vary perfectly if not all lanes are filled) one could account for – and test - different theories.

Response: Thanks for these great comments. The evidence from the baseline specification is simply showing the net impact of being in a specific lane. I think this is ultimately what we care about in the track and field context, i.e. “are the middle lanes best?” But there is a subtle point you raise that the “being behind spurs effort” channel could still be active, but it’s not strong enough to make the outside lanes slower. This question is more relevant for the motivational/psychological factors you highlight. Following your suggestion, I’ve added a separate analysis where I include an indicator variable for the outmost lane (which, as you note, is not perfectly correlated with lane number). I pursue this additional analysis in the 200 and 400m, where the staggered starts/effort effects would be most noticeable and find some evidence that the outermost lanes in the 200m do slow runners down. I find no statistical effect of the outermost lane in the 400m, which could be a product of noisier data. Anyway, thanks for these suggestions, I think they have really strengthened the paper.

- I was kind of surprised that wind but not wind shadow created by opponents – where lane assignment is probably relevant – was discussed.

Response: Thanks for this comment. I’ve never encountered it anecdotally from my participation in the sport and I searched for any discussion in track and field forums about wind shadows in sprint events and was unable to find anything (it’s certainly a factor in long- distance events where runners draft, but I couldn’t find anything in relation to sprinting). So, it seems like it’s not a common thing that is highlighted in relation to lane assignments. I worry about adding discussion about this since it doesn’t seem to be widely discussed, so I have left it out.

- What if the average skill athletes’ level in the first round of these tournaments is not high enough that the finer features of the biomechanics and other subtle factors affecting performance just don’t play out to be decisive? Maybe even the first round is highly challenging to get into and only achievable for pros, but without further contextual info that’s hard to judge.

Response: I think this is an interesting question. My off-the-cuff response is that these are the world championships, so clearly these are the best runners in the world. I’ve tried to think of different ways to provide more contextual info, but nothing obvious came to mind... (e.g. look at average season’s bests of this group of runners, but compare it to who?) Being the world championships doesn’t necessarily rule out that these lane effects are only/more salient for the cream of the crop, but I don’t know if there is a way to assess that question reliably. Ultimately, exploiting the random assignment feature is important for obvious reasons, and this feature doesn’t exist in the more elite rounds (e.g. semis/finals). I don’t disagree with the possibility that lane effects would be more relevant for more elite athletes, but without a reliable way to empirically assess the question it feels very conjecture-y, so I have left this discussion out.

Econometric specification and presentation

- Are 8 lane dummies really the best specification, especially when there are concerns about statistical power? If effects are expected to be monotone across lanes, why not a continuous (linear or quadratic) function?

Response: See below.

- If, there is a good reason to not use a continuous specification, why can lanes not be grouped (e.g., into 1&2, 3&4, 5&6, 7&8) to achieve more power per coefficient?

Response: Thanks for these comments. In the baseline analysis I chose dummies for each lane to be as agnostic as possible regarding the functional form of any lane effects. As additional robustness checks, and to improve statistical power, I have added additional results where I group 1&2, 3&4, 5&6, 7&8&9 and where I treat lanes as a continuous variable. The general conclusions from the original baseline specification are also reflected in these additional results.

- Reading essentially 5 times the same specification and table but with different data sets is kind of hard and makes it hard to compare findings. Why not only present the 100m table as an example in the main text and then depict the (relevant) line coefficient and their SE graphically (e.g., lanes on the x-axis, normalised coefficients on the y-axis with SE and a line connecting them). This graphical presentation could then be added for all other distance to the same graph (e.g. by stacking the connecting lines with the coefficients). All other tables and the randomness check can then go to an appendix and the paper would be much more comprehensive. (Ideally the graph would also depict a normalised interval for MDE for each line.)

Response: I agree that there are a lot of tables to look at. I’ve moved the randomness checks for the 200, 400, and 800m to the appendix. I’ve also added a graphical depiction of the main results. Stacking everything in one graph became visually very cluttered. So, what I’ve done is to graphically display the estimated lane effects (from pooling men and women and excluding outliers) together with 95% confidence intervals in side-by-side figures for the 100, 200, 400 and 800m. I think this graphical display helps to interpret the results. I’ve left the regression result tables in the main text as I refer to them frequently in the text and it would be burdensome for readers to flip back and forth between the appendix.

Other (some minor, some not) issues

- “was put in place during in the 1985-86 rules under rule 141.11” reads like the rule were only in place in the years/season 1985/1986 while elsewhere we find “AAF World Championships and U20 World Championships from 2000 to 2019”. I suggest to clarify this (also footnote 3 which is hard to read).

- In general, I would advise to describe a bit more clearly how/when/how long the random assignment was introduced. Right now it reads just like it was and then a bunch of technical details. Maybe that can be presented more story-like.

Response: I have re-worded the section to emphasize that the random assignment rule was initiated in the 1985-86 season and remains in place today. I wish I had more of a “rule history” (i.e. why were rules changed, etc.) to rely on to tell a story here, but unfortunately I’ve been unable to find that information anywhere.

- I would also advise to refer to measures in the flow of the text not by their variable names but by what they describe in order to improve readability.

Response: I’ve done this where appropriate.

- the variable SB is first described as “SB is the runner’s season best race time”, then as “ assigned lanes based on prior race results (proxied here by SB)”. Am I correct that the author tries to claim that results prior to race can be proxied by SB, e.g., the best result across the whole season?

Response: I have re-written this section to help clarify.

- The sum in the regression equation should index over only 8 dummies, not 9, as one lane is the baseline.

Response: thanks for point this out, I have made this change.

- I did not find the data in the paper or an appendix, except for a link to a sports website. I would expect the authors to share, with the manuscript, i) the original dataset used, including any outliers or incomplete data dropped for the actual analysis, ii) a short description on how it was generated, iii) the script(s) used to analyse and pre-process the data and to generate all tables/figures.

Response: My apologies for this. I’m used to providing replication packages during the publication process, not prior. While the data is publicly available in uncompiled form from the IAAF, I have now included a link to a replication package with the compiled data.

Reviewer 2:

The topic of the paper is undoubtedly interesting and appealing. However, I have very serious concerns on the validity of the results presented in the paper due to the very poor and

inaccurate description of the applied statistical methodology. Following, the details of the review are divided into major and minor comments.

Major comments:

1) The general description of the statistical methodology applied in the paper is completely missing. This issue does not allow to appropriately evaluate the validity of the results reported in the paper. The author should to carefully describe the applied statistical methodology in details in a separate section, by reporting in a rigorous way the main theory (formulas, assumptions, and the corresponding references). Furthermore, the specifications of the statistical models in formulas (1) and (2) are completely inaccurate; for instance, subscripts are missing in these formulas, the random components are just reported as “error” rather than through the well-known statistical notation, models’ assumptions are completely missing.

Response: My apologies for this omission. In the field I’m in, it would be seen as unnecessarily/obvious to discuss the theory/assumptions underlying the regression analysis. With random assignment to treatment, the analysis is quite straight and amounts to reporting average treatment effects. To be as agnostic as possible about the structure of these effects I estimate them using dummy (0,1) variables for lanes in a regression that controls for the other covariates. I have added some discussion of this to clarify.

2) My main concern relates to the validity of the results reported in the paper. Since the description of the statistical methodology is completely missing, from what I see, it seems that the author apply a linear regression model? At the same time, the author claims along the entire manuscript to estimate “the causal effect of line assignments on race times”. However, it is well-known that we cannot speak about a causal effect when considering the “classical” linear regression model. There are several specific approaches for causal inference, as for instance the potential outcomes framework, causal graphs and similar; however, nothing is mentioned in the paper on this point. The issue mentioned above is of crucial importance on the entire validity and interpretation of the results reported in this paper, and it should be carefully justified and explained in details.

Response: Respectfully, “However, it is well-known that we cannot speak about a causal effect when considering the “classical” linear regression model” is simply incorrect. There is nothing inherently problematic about using linear regression to estimate causal effects. The issue is whether the assumptions of OLS are met or not. In non-experimental data, it is unlikely that the assumptions are met. But the whole point of the paper is to leverage random assignment to ensure the exogeneity assumption is met. There is nothing wrong with using OLS to estimate average treatment effects with random assignment.

3) Statistical model diagnostics are completely missing. They should be performed and reported in order to appropriately evaluate the estimated statistical models.

Response: As is now clarified in the paper, there is no functional form assumed in estimating the average treatment effects of lanes. As a result, in terms of model diagnostics, there is nothing to test regarding the functional form or homoscedasticity for the main variables of interest (lane effects). In response to the other reviewer’s suggestions, I have also added results estimated from two different regression specifications.

4) Tables no.1-no.8: in all the tables, the estimated coefficients are reported incorrectly as Lane 1, Lane 3, Wind, etc. For example, the author should to write β _̂ 1 rather than Lane 1, β _̂ 3 rather than Lane 3, α _̂ 1 rather than Wind, and so on. Moreover, R^2 are erroneously reported as “R ^ 2”, and furthermore they should be discussed.

Response: Thanks for this comment. I have raised “2” to a superscript in “R^2”. For readability sake, I feel “Lane 1” is preferrable to “\\beta_1”. It seems very straightforward to understand that “Lane 1” in these tables is referring the coefficient estimates for Lane 1.

Minor comments:

1) To the best of my knowledge, the PLOS One Guidelines for authors require that the references in the text are reported by numbers, and footnotes are not permitted. Please, correct.

Response: thank you for pointing this out, I have made these changes.

---

## [Decision Letter · Decision Letter 1]

14 Apr 2022

PONE-D-21-35820R1Are there lane advantages in track and field?PLOS ONE

Dear Dr. Munro,

Thank you for submitting your manuscript to PLOS ONE. After careful consideration, we feel that it has merit but does not fully meet PLOS ONE’s publication criteria as it currently stands. Therefore, we invite you to submit a revised version of the manuscript that addresses the points raised during the review process.

We look forward to receiving your revised manuscript.

Kind regards,

Roy Cerqueti, Ph.D.

Academic Editor

PLOS ONE

Additional Editor Comments (if provided):

Reviewers' comments:

Reviewer's Responses to Questions

**Comments to the Author**

1. If the authors have adequately addressed your comments raised in a previous round of review and you feel that this manuscript is now acceptable for publication, you may indicate that here to bypass the “Comments to the Author” section, enter your conflict of interest statement in the “Confidential to Editor” section, and submit your "Accept" recommendation.

Reviewer #2: (No Response)

Reviewer #3: (No Response)

2. Is the manuscript technically sound, and do the data support the conclusions?

Reviewer #2: No

Reviewer #3: Yes

3. Has the statistical analysis been performed appropriately and rigorously? 

Reviewer #2: No

Reviewer #3: Yes

4. Have the authors made all data underlying the findings in their manuscript fully available?

Reviewer #2: Yes

Reviewer #3: Yes

5. Is the manuscript presented in an intelligible fashion and written in standard English?

Reviewer #2: Yes

Reviewer #3: Yes

6. Review Comments to the Author

Reviewer #2: As already stated in my previous report, the topic of the paper is undoubtedly interesting and appealing. However, I still remain with the concerns included in my previous report, mainly due to the fact that almost all major comments/suggestions I made to improve the paper are not taken into account. The statistical methodology applied in the paper is still not clearly described in a general and rigorous way; the statistical models are not reported accurately, for instance formulas (1) and (2). From some author’s clarifications, I was able to understand that the author deals with a causal inference framework, but its general and clear description is missing, as well as the relevant literature in this field. Following, the details of the review is reported into major comments.

Major comments:

1. The general description of the statistical methodology applied in the paper is still completely missing. The point is that this cause a misunderstanding of the applied statistical methodology, and it does not allow to appropriately evaluate the validity of the results reported in the paper. From what I was able to understand from the author’s corrections made in the manuscript, and some responses, the author deals with a causal inference framework, rather than with classical linear regression model. Regarding on the field one works, in my opinion, the causal inference framework the author deals with, is anything but “unnecessarily/obvious to discuss the theory/assumptions underlying the regression analysis.” I wish also to point out that it is not just a regression analysis, but a causal inference framework which is something different from classical linear regression modelling. I still suggest to the author to carefully describe it in details in a separate section, by reporting in a rigorous way the main theory (formulas, assumptions, and also the most relevant literature in the causal inference framework which is completely missing). Regarding the statistical models in formula (1) and (2), they are still reported inaccurately: 〖Time〗_(i,j) to indicate the response variable? Statistical models should be reported accurately in general, for instance using y_ij for the response variable. What about the subscripts: i=1,…?,j=1,…? Also, some statistical terminology: for instance, along the paper just using “specification” along the paper to refer to a statistical model?

2. Again, it is well-known that we cannot speak about a causal effect when considering the “classical” linear regression model. Correlation does not imply causality! Using OLS regression to estimate average treatment effects with random assignment is a causal inference framework, not a “classical” linear regression modelling. This is also why I kindly invite the author to describe briefly describe in a rigorous way the applied statistical methodology, in order to make the paper clear for potential readers.

3. Being revealed that the author use a causal inference framework, and not just a classical linear regression modelling, I have a question. More precisely, the random assignment assumption is crucial for the validity of the results. The author check such assumption through the statistical model in formula (2): do you think that this is enough to confirm it, and why? What about existing approaches in the literature to deal with this issue; for instance, just to mention one (i.e., not limited to), the propensity score approach? Furthermore, in Section 3.1, in the sentence “…only lane 1 has a statistically significant relationship with SB, which, again, is likely an effect of the low number of observations.” Why it should be due to “an effect of the low number of observations”?

4. Regarding the results for the “pooled” data: why a covariate for gender is not included in the statistical models? How results change if you include also “gender” as a covariate in an appropriate way?

5. Again, in my opinion, suitable models diagnostics should be performed in order to appropriately evaluate the estimated statistical models. They are not just limited to evaluate the “functional form”. Just a clarification: the homoscedasticity assumption relates to the error component, and not to the “main variables of interest (lane effects)”.

6. Again, Tables no.1-no.8: in all the tables, the estimated coefficients are reported incorrectly as Lane 1, Lane 3, Wind, etc. For example, the author should to write β ^_1 rather than Lane 1, β ^_3 rather than Lane 3, α ^_1 rather than Wind, and so on. It could seem very straightforward to understand, but the problem is that this is not correct, and it’s an error.

Reviewer #3: The paper focuses on the common belief about the fact that some lanes on the track, in particular the middle ones such as 3-6, are advantageous with respect to the others. Using a sample of random assigned lanes in the first round of events the author finds no evidence supporting the common belief and, in some cases, even contrary. The work concludes that the common belief is a folk tale.

General comments

The paper answers an interesting and “fanciful” question about an alleged benefit in racing in the middle lanes. The paper is well motivated and the results are likely to have a great echo in non-academic fields. The econometric analysis is correctly conducted, and the results obtained seem sound. Notwithstanding, there are some points that deserve to be refined. In particular:

1) notation of eq (1) and (2). As far as I’ve understood, the dummies Lk represent the Lanek for runner i in heat j and their value is supposed to change according to the runner and to the heat. Therefore, they should also report the subscript ij and the summation is up to k-1 given the collinearity problem.

2) When the pulled sample is used a gender dummy should be included

3) Quoting from page 7 “only lane 1 has a statistically significant relationship with SB, which, again, is likely an effect of the low number of observations”. It is difficult to understand this claim, it seems the other way round. When the number of observations increases, the standard error (SE) decreases, and consequently the t-stat increases as t=beta/SE(beta), therefore when the number of observations is low one over accepts the null of non significant statistical effects, i.e. weak power. I think the claim should be revised and another piece of explanation for that significance should be put forth.

4) Power problems. I wonder whether there are other tests that can be implemented to analyse the issue whether the results are driven by low power or whether they can be read as pure lack of statistical significance. I am not an expert of this specific filed, but one idea could be to adapt the tests proposed by Cattaneo Titiunik and Vazquez-Bera (2019) to the case under scrutiny.

Minor issues

The references must be reported according to the common practice followed in the literature. In the current version, no contribution reports the year.

References

Cattaneo M-D., Titiunik R., and Vazquez-Bera G., 2019. Power calculations for regression-discontinuity designs. The Stata Journal, 19(1): 210-245.

7. PLOS authors have the option to publish the peer review history of their article (what does this mean?). If published, this will include your full peer review and any attached files.

Reviewer #2: No

Reviewer #3: No

---

## [Author Response · Author response to Decision Letter 1]

18 May 2022

Response to Reviewers:

Reviewer #1

Thank you for your positive assessment of my original manuscript and helpful comments. I hope my responses and changes to the manuscript were satisfactory.

Reviewer #3

Thank you for your time and effort in reviewing my manuscript along with your helpful suggestions. Below I’ve responded to each of your comments and detailed the changes made to the paper in bold below each.

The paper answers an interesting and “fanciful” question about an alleged benefit in racing in the middle lanes. The paper is well motivated and the results are likely to have a great echo in non-academic fields. The econometric analysis is correctly conducted, and the results obtained seem sound. Notwithstanding, there are some points that deserve to be refined. In particular:

1) notation of eq (1) and (2). As far as I’ve understood, the dummies L_k represent the Lane k for runner i in heat j and their value is supposed to change according to the runner and to the heat. Therefore, they should also report the subscript ij and the summation is up to k-1 given the collinearity problem.

Response: Thank you for pointing this out, I have made these suggested changes. I modified the summation up to n-1, as the notation should be distinct from the index variable k.

2) When the pulled sample is used a gender dummy should be included

Response: Thanks for this suggestion. A gender dummy wasn’t included originally because Personal and Seasonal Bests should pick up a lot of the gender differences. But there certainly could be added information controlled for with a gender dummy, so I have included it in the pooled regressions. The general conclusions from the regressions don’t change. If anything, the dummy tightens up some of the standard errors in the pooled regressions, and some of the results are slightly stronger. Again, thanks for this suggestion.

3) Quoting from page 7 “only lane 1 has a statistically significant relationship with SB, which, again, is likely an effect of the low number of observations”. It is difficult to understand this claim, it seems the other way round. When the number of observations increases, the standard error (SE) decreases, and consequently the t-stat increases as t=beta/SE(beta), therefore when the number of observations is low one over accepts the null of non significant statistical effects, i.e. weak power. I think the claim should be revised and another piece of explanation for that significance should be put forth.

Response: Thanks for these comments. The idea is that with low N, it’s more likely to have Type-I error (erroneously rejecting the null of no effect). I have added a note about this and a reference to Leppink et al. 2016. The pooled results for the randomization checks have changed to some degree with the gender dummy, so lane 1 is no longer significant. In the 100m, only lane 4 is significant, however, with multiple lanes/hypothesis tests, a more appropriate test is to examine if the lane assignments are jointly significant. As such, to strengthen these randomization checks, I have also added F-tests to examine the joint significance of the lanes, and by and large they fail significance at 5% level. In only the women’s 100m, F-tests are statistically significant, but they fail 5% significance when the data is pooled with the men’s results. In the 200, 400 and 800m all the F-test results are highly insignificant. These results suggest that, collectively, the lane assignments are unrelated to the prior performance of runners, i.e. randomly assigned.

4) Power problems. I wonder whether there are other tests that can be implemented to analyse the issue whether the results are driven by low power or whether they can be read as pure lack of statistical significance. I am not an expert of this specific filed, but one idea could be to adapt the tests proposed by Cattaneo Titiunik and Vazquez-Bera (2019) to the case under scrutiny.

Response: Thanks for this suggestion. I looked at the suggested paper, and it’s tailored to regression discontinuity research designs, which, unfortunately, are a different identification strategy than I take. I’m not an expert with power calculations, but I did some more research and it seems like reporting MDEs, which I do in the paper, is a common approach to discussing ex-post statistical power (e.g. Mckenzie and Ozier 2019). I have added more discussion and a citation motivating the use of MDEs.

Minor issues

The references must be reported according to the common practice followed in the literature. In the current version, no contribution reports the year.

Response: Reporting the years is also the practice I’m familiar and comfortable with, however the PLOS ONE guidelines state: “References are listed at the end of the manuscript and numbered in the order that they appear in the text. In the text, cite the reference number in square brackets (e.g., “We used the techniques developed by our colleagues [19] to analyze the data”).”

References

Cattaneo M-D., Titiunik R., and Vazquez-Bera G., 2019. Power calculations for regression-discontinuity designs. The Stata Journal, 19(1): 210-245.

References:

Leppink, Jimmie, Kal Winston, and Patricia O’Sullivan. "Statistical significance does not imply a real effect." Perspectives on medical education 5.2 (2016): 122-124.

McKenzie, David, and Owen Ozier. "Why ex-post power using estimated effect sizes is bad, but an ex-post MDE is not." World Bank Development Impact Blog (2019).

Reviewer #2: 

I appreciate your time and effort in reviewing my manuscript. Please see my responses to your comments below.

As already stated in my previous report, the topic of the paper is undoubtedly interesting and appealing. However, I still remain with the concerns included in my previous report, mainly due to the fact that almost all major comments/suggestions I made to improve the paper are not taken into account. The statistical methodology applied in the paper is still not clearly described in a general and rigorous way; the statistical models are not reported accurately, for instance formulas (1) and (2). From some author’s clarifications, I was able to understand that the author deals with a causal inference framework, but its general and clear description is missing, as well as the relevant literature in this field. Following, the details of the review is reported into major comments.

Major comments:

1. The general description of the statistical methodology applied in the paper is still completely missing. The point is that this cause a misunderstanding of the applied statistical methodology, and it does not allow to appropriately evaluate the validity of the results reported in the paper. From what I was able to understand from the author’s corrections made in the manuscript, and some responses, the author deals with a causal inference framework, rather than with classical linear regression model. Regarding on the field one works, in my opinion, the causal inference framework the author deals with, is anything but “unnecessarily/obvious to discuss the theory/assumptions underlying the regression analysis.” I wish also to point out that it is not just a regression analysis, but a causal inference framework which is something different from classical linear regression modelling. I still suggest to the author to carefully describe it in details in a separate section, by reporting in a rigorous way the main theory (formulas, assumptions, and also the most relevant literature in the causal inference framework which is completely missing). Regarding the statistical models in formula (1) and (2), they are still reported inaccurately: 〖Time〗_(i,j) to indicate the response variable? Statistical models should be reported accurately in general, for instance using y_ij for the response variable. What about the subscripts: i=1,…?,j=1,…? Also, some statistical terminology: for instance, along the paper just using “specification” along the paper to refer to a statistical model?

Response: Thanks for these comments. I had a hard time interpreting what is being requested here as there are no specific references to relevant literature. The identification strategy is random assignment to treatment. In this case, understanding causal effects amounts to reported average treatment effects. While I implement a regression-based approach, all the regressions are doing is computing the difference in average times by lane number. One could just compute raw means and do this, but the regression framework is convenient because it allows me to control for other correlates, which helps improve precision. In an attempt to respond to the request for more discussion of causal inference, I have added some discussion about what the random assignment buys you (i.e. the equivalence of characteristics of the treatment groups). Thank you for the suggestion regarding subscripts. Following your suggestion and one from the other reviewer I have made some modifications to the notation. I have also changed the language to “regression specification” to avoid any confusion.

2. Again, it is well-known that we cannot speak about a causal effect when considering the “classical” linear regression model. Correlation does not imply causality! Using OLS regression to estimate average treatment effects with random assignment is a causal inference framework, not a “classical” linear regression modelling. This is also why I kindly invite the author to describe briefly describe in a rigorous way the applied statistical methodology, in order to make the paper clear for potential readers.

Response: Thanks for these comments. However, I respectfully disagree. The word “classical” simply refers to the case when the assumptions of OLS are met. There is nothing inherently wrong with using regression for causal inference *if* you are confident that assignment to treatment is random. This is the whole point of the paper-- to leverage the random assignment to lanes to estimate a causal effect. In other words, OLS is simply being used as a statistical method to test a null hypothesis of differences in the data that are generated from random variation. For more discussion on using regression to estimate causal treatment effects with random assignment see Ch. 9 of Gelman and Hill (2006). I have added some discussion about random assignment and causal inference to help clarify the identification strategy. 

3. Being revealed that the author use a causal inference framework, and not just a classical linear regression modelling, I have a question. More precisely, the random assignment assumption is crucial for the validity of the results. The author check such assumption through the statistical model in formula (2): do you think that this is enough to confirm it, and why? What about existing approaches in the literature to deal with this issue; for instance, just to mention one (i.e., not limited to), the propensity score approach? Furthermore, in Section 3.1, in the sentence “…only lane 1 has a statistically significant relationship with SB, which, again, is likely an effect of the low number of observations.” Why it should be due to “an effect of the low number of observations”?

Response: Thanks for these comments. I’m sympathetic to the concern about random assignment to treatment, this is an important assumption. In terms of why we should believe it: as stated in the paper, it is in the competition rulebook of the IAAF. It is, of course, possible that they don’t follow the rules, so the randomization checks were done as an attempt to confirm adherence to this rule, and they are generally supportive. To conduct them, I look at a runners Season’s Best listed in the startlist for each heat. This is the only observable information on the runners I have available (besides gender, but I also group gender separately). Propensity score matching is useful when you are exploring differences across a number of characteristics. But, unfortunately, I only have one characteristic. For the question at hand though -- how race times vary by lane -- SB is arguably the most relevant characteristic as it proxies very well for a runner’s ability. To strengthen the randomization check results I have also added F-tests to examine the joint significance of the lanes. In terms of the low number of observations, the concern is that with smaller sample sizes it’s more likely to have Type-1 error (incorrectly rejecting the null of no effect). I have added a note about this and a reference to Leppink et al. 2016.

4. Regarding the results for the “pooled” data: why a covariate for gender is not included in the statistical models? How results change if you include also “gender” as a covariate in an appropriate way?

Response: Thanks for this comment, I have added gender as a control in the pooled regressions.

5. Again, in my opinion, suitable models diagnostics should be performed in order to appropriately evaluate the estimated statistical models. They are not just limited to evaluate the “functional form”. Just a clarification: the homoscedasticity assumption relates to the error component, and not to the “main variables of interest (lane effects)”.

Response: Thanks for these comments. As noted in the results tables, the standard errors reported are all heteroscedasticity-consistent (i.e. robust standard errors). 

6. Again, Tables no.1-no.8: in all the tables, the estimated coefficients are reported incorrectly as Lane 1, Lane 3, Wind, etc. For example, the author should to write β ^_1 rather than Lane 1, β ^_3 rather than Lane 3, α ^_1 rather than Wind, and so on. It could seem very straightforward to understand, but the problem is that this is not correct, and it’s an error.

Response: Thanks for these comments. I’m sympathetic to your point, yet the norm is to label the coefficients estimates with the names of independent variables, and not, e.g., \\beta_1. Indeed, this is how statistical software (e.g. R, Stata, etc.) reports regression results. I think it’s important to follow the norm, so I have left the labelling as is. This practice also avoids the need for readers to keep referring back and forth between to the regression specification and the results tables to understand what each \\beta represents.

References:

Gelman, Andrew, and Jennifer Hill. Data analysis using regression and multilevel/hierarchical models. Cambridge university press, 2006.

---

## [Decision Letter · Decision Letter 2]

17 Jun 2022

PONE-D-21-35820R2Are there lane advantages in track and field?PLOS ONE

Dear Dr. Munro,

Thank you for submitting your manuscript to PLOS ONE. After careful consideration, we feel that it has merit but does not fully meet PLOS ONE’s publication criteria as it currently stands. Therefore, we invite you to submit a revised version of the manuscript that addresses the points raised during the review process.

ACADEMIC EDITOR: Dear David, one reviewer accepts the paper, while the other one raises criticisms on some crucial aspects of the study. Please, provide a detailed response to the points raised by the reviewer. I want to stress that only a satisfactory treatment of such points might lead to the acceptance of the paper.Thank you for your effort, I wish you all the best.Yours,Roy

We look forward to receiving your revised manuscript.

Kind regards,

Roy Cerqueti, Ph.D.

Academic Editor

PLOS ONE

Reviewers' comments:

Reviewer's Responses to Questions

**Comments to the Author**

1. If the authors have adequately addressed your comments raised in a previous round of review and you feel that this manuscript is now acceptable for publication, you may indicate that here to bypass the “Comments to the Author” section, enter your conflict of interest statement in the “Confidential to Editor” section, and submit your "Accept" recommendation.

Reviewer #2: (No Response)

Reviewer #3: (No Response)

2. Is the manuscript technically sound, and do the data support the conclusions?

Reviewer #2: (No Response)

Reviewer #3: Yes

3. Has the statistical analysis been performed appropriately and rigorously? 

Reviewer #2: (No Response)

Reviewer #3: Yes

4. Have the authors made all data underlying the findings in their manuscript fully available?

Reviewer #2: Yes

Reviewer #3: Yes

5. Is the manuscript presented in an intelligible fashion and written in standard English?

Reviewer #2: Yes

Reviewer #3: Yes

6. Review Comments to the Author

Reviewer #2: The topic of the paper is undoubtedly appealing and interesting. However, in this third round of revision, my concerns remain the same, since most of my major comments/suggestions to improve the paper are again not taken fully and/or appropriately into account. To this end, in what follows, once again I report almost the same major comments present in my two previous reports.

Major comments:

1. The general description of the statistical methodology applied in the paper, as well as the relevant literature in this field are still completely missing. I still kindly suggest to the author to carefully describe in details in a separate section and/or subsection the main statistical theory (i.e., the statistical methodology used in this paper) in a general and rigorous way, i.e., main formulas, assumptions and so on. Moreover, the most relevant literature in the causal inference framework, which is still completely missing, should be also reported by the author and cited where necessary. Regarding the statistical models in formula (1) and (2), they are still reported inaccurately: again, why 〖Time〗_(i,j) is used to indicate the response variable? Statistical models should be reported accurately in general, for instance using y_ij for the response variable. Similar issues also apply to the independent variables included in the model. Again, what about the subscripts: i=1,…?,j=1,…? That is, the subscripts “i” and “j” goes from 1 to what? About the subscript “k”: why it goes from 1 to n? Why not to “K” for example, since the letter “n” in statistics is usually used to indicate the sample size. Also, I think it is confused and not in line with the basic statistical terminology to use always and everywhere “regression specification” only to refer to a statistical model.

2. In fact, it is well-known that in a causal inference framework with random assignment to treatment, one can use a regression model to estimate causal effects. But, please note that when one deals with classical linear regression modelling (NOT in a causal inference framework with the well-known assumptions on the treatment assignments mechanism), one cannot state that this is a causal effect, because correlation does not imply causality. I understood that the author deals with causal inference framework, but in my opinion, it should be accurately described in a separate section and/or subsection through a general description of the main statistical theory, as I already suggested at my previous major comment #1.

3. Please, justify and explain better your statement on the use of propensity score matching in your specific case-study. Furthermore, my previous question was not just limited to propensity score matching: what about other existing approaches in the literature to deal with this issue? The statements along the paper which rely on “low number of observations” are still very confused, and, at least in this current form, they do not seem appropriate. This is because from what I see in all the tables, the values reported for “N” are high rather than low. Please, justify.

4. Again, in my opinion, suitable models diagnostics should be performed in order to appropriately evaluate the estimated statistical models. Please, note that they are definitely not just limited to standard errors.

5. Again, Tables no.1-no.8: in all the tables, the estimated coefficients are reported incorrectly as Lane 1, Lane 3, Wind, etc. For example, the author should to write β ^_1 rather than Lane 1, β ^_3 rather than Lane 3, α ^_1 rather than Wind, and so on. It could seem very straightforward to understand, it is how R, Stata reports them, but the problem is that this is not correct, it’s not the norm and it’s an error.

Reviewer #3: (No Response)

7. PLOS authors have the option to publish the peer review history of their article (what does this mean?). If published, this will include your full peer review and any attached files.

Reviewer #2: No

Reviewer #3: No

---

## [Author Response · Author response to Decision Letter 2]

1 Jul 2022

Response to Reviewers:

Reviewer #2: The topic of the paper is undoubtedly appealing and interesting. However, in this third round of revision, my concerns remain the same, since most of my major comments/suggestions to improve the paper are again not taken fully and/or appropriately into account. To this end, in what follows, once again I report almost the same major comments present in my two previous reports.

Major comments:

1. The general description of the statistical methodology applied in the paper, as well as the relevant literature in this field are still completely missing. I still kindly suggest to the author to carefully describe in details in a separate section and/or subsection the main statistical theory (i.e., the statistical methodology used in this paper) in a general and rigorous way, i.e., main formulas, assumptions and so on. Moreover, the most relevant literature in the causal inference framework, which is still completely missing, should be also reported by the author and cited where necessary. Regarding the statistical models in formula (1) and (2), they are still reported inaccurately: again, why 〖Time〗_(i,j) is used to indicate the response variable? Statistical models should be reported accurately in general, for instance using y_ij for the response variable. Similar issues also apply to the independent variables included in the model. Again, what about the subscripts: i=1,…?,j=1,…? That is, the subscripts “i” and “j” goes from 1 to what? About the subscript “k”: why it goes from 1 to n? Why not to “K” for example, since the letter “n” in statistics is usually used to indicate the sample size. Also, I think it is confused and not in line with the basic statistical terminology to use always and everywhere “regression specification” only to refer to a statistical model.

Response: see response to 2.

2. In fact, it is well-known that in a causal inference framework with random assignment to treatment, one can use a regression model to estimate causal effects. But, please note that when one deals with classical linear regression modelling (NOT in a causal inference framework with the well-known assumptions on the treatment assignments mechanism), one cannot state that this is a causal effect, because correlation does not imply causality. I understood that the author deals with causal inference framework, but in my opinion, it should be accurately described in a separate section and/or subsection through a general description of the main statistical theory, as I already suggested at my previous major comment #1.

Response: I agree that without random assignment linear regression can’t speak about causal effects. In relation to your comments 1 and 2 I have added a separate section detailing the causal inference framework, which highlights the importance of the independence assumption (treatment status independent of outcomes). This emphasizes the importance of utilizing the heats which implement random assignment to lanes. 

As per your request, I have changed the notation to Y_{i,j} instead of “Time.” The i and j notation is not over sums, so they are just denoting unique observations for different “i” and “j”, so it is not necessary to stipulate the limits of these indexes. In relation to the 1 to n notation, I changed the notation in rewriting the causal inference section and no longer use “n.”

As per your request, I have also changed “specification” to “model.” 

3. Please, justify and explain better your statement on the use of propensity score matching in your specific case-study. Furthermore, my previous question was not just limited to propensity score matching: what about other existing approaches in the literature to deal with this issue? The statements along the paper which rely on “low number of observations” are still very confused, and, at least in this current form, they do not seem appropriate. This is because from what I see in all the tables, the values reported for “N” are high rather than low. Please, justify.

Response: I believe your comment is referring to propensity score matching in relation to the randomization checks. Normally, propensity score matching is a way to match individuals in the treatment and control groups based on similar covariates (we can reduce the dimensionality of this by summarizing similar individuals by their propensity scores.) This matching is typically done when there are concerns that individuals in the treatment/control groups are systematically different. In this sense, the matching part only really makes sense when propensities vary systematically conditional on treatment status. However, one can use the first step (estimating propensities) to see if the randomization successfully balanced treatment and control groups. I think this is what you may be asking for. In the appendix I have added probit regressions where I estimate how the probability of being assigned to a lane is a function of runner ability (season’s best). If these treatment probabilities vary systematically by runner ability that would be concerning about the randomization. Thankfully, none of the regressions show that season’s best is significantly related to treatment status. I hope I have interpreted your comment appropriately. 

In relation to the sample sizes, what I’m trying to highlight is that readers should be cautious about any statistical significance being derived from small sample sizes. For example, in the Women’s 100m randomization check, there are statistically significant results for lanes 1 and 9. However, the sample sizes in these lanes are less than 40, relative to around 100 in the other lanes. These are relatively small sample sizes. The concern is that with smaller sample sizes it’s more likely to have Type-1 error (incorrectly rejecting the null of no effect) (Leppink et al. 2016.). This also relates to Type-M error (Gelman and Carlin, 2014). They emphasize that significant results from small sample sizes often overstate the magnitude of the true effect: “The problem, though, is that if sample size is too small, in relation to the true effect size, then what appears to be a win (statistical significance) may really be a loss (in the form of a claim that does not replicate).” My point is that in lanes with small numbers of observations, while there are occasionally significant results, readers should be skeptical of the replicability of those results. I have attempted to clarify this in the text. 

4. Again, in my opinion, suitable models diagnostics should be performed in order to appropriately evaluate the estimated statistical models. Please, note that they are definitely not just limited to standard errors.

Response: There are several diagnostics/robustness checks related to the model already in the paper: R^2, F-statistics, removing outliers, robust standard errors, and three different statistical models. None of the results are sensitive to these different checks. It’s also important to emphasize that the coefficients of interest are on indicator variables (treatment indicators), and there is no functional form assumed here. It may also be relevant to note that with random assignment, choosing the appropriate model is not really necessary for a causal interpretation of regression. For more discussion of this, see Chp. 3 in Angrist and Pischke (2009). 

Without specific guidance on what diagnostic(s) you think would be valuable to add and why, I don’t know how else to respond to this comment. I apologize.

5. Again, Tables no.1-no.8: in all the tables, the estimated coefficients are reported incorrectly as Lane 1, Lane 3, Wind, etc. For example, the author should to write β ^_1 rather than Lane 1, β ^_3 rather than Lane 3, α ^_1 rather than Wind, and so on. It could seem very straightforward to understand, it is how R, Stata reports them, but the problem is that this is not correct, it’s not the norm and it’s an error.

Response: I’ve changed the notation in the tables to be \\beta_1 etc., as you have requested. However, as a search of papers in PLOS ONE will highlight, I do not think this is a standard practice when reporting regression results, at least in the fields I am familiar with. To ease interpretation in light of this change, I have also included the independent variable in parenthesis beside each coefficient, so readers know what each coefficient represents. I hope you find this to be a reasonable compromise.

---

## [Editor Report · Decision Letter 3]

6 Jul 2022

Are there lane advantages in track and field?

PONE-D-21-35820R3

Dear Dr. Munro,

We’re pleased to inform you that your manuscript has been judged scientifically suitable for publication and will be formally accepted for publication once it meets all outstanding technical requirements.

Kind regards,

Roy Cerqueti, Ph.D.

Academic Editor

PLOS ONE

Additional Editor Comments (optional):

Dear David,

I'm satisfied with your revision strategy, No further revision rounds are required by my side.

Thanks a lot, yours,

Roy